# Mice in a labyrinth show rapid learning, sudden insight, and efficient exploration

**Matthew Rosenberg[1†], Tony Zhang[1†], Pietro Perona[2]\*, Markus Meister[1]\***

[1]Division of Biology and Biological Engineering, California Institute of Technology, Pasadena, United States; [2]Division of Engineering and Applied Science, California Institute of Technology, Pasadena, United States

**Abstract** Animals learn certain complex tasks remarkably fast, sometimes after a single experience. What behavioral algorithms support this efficiency? Many contemporary studies based on two-alternative-forced-choice (2AFC) tasks observe only slow or incomplete learning. As an alternative, we study the unconstrained behavior of mice in a complex labyrinth and measure the dynamics of learning and the behaviors that enable it. A mouse in the labyrinth makes ~2000 navigation decisions per hour. The animal explores the maze, quickly discovers the location of a reward, and executes correct 10-bit choices after only 10 reward experiences — a learning rate 1000-fold higher than in 2AFC experiments. Many mice improve discontinuously from one minute to the next, suggesting moments of sudden insight about the structure of the labyrinth. The underlying search algorithm does not require a global memory of places visited and is largely explained by purely local turning rules.

## Introduction

How can animals or machines acquire the ability for complex behaviors from one or a few experiences? Canonical examples include language learning in children, where new words are learned after just a few instances of their use, or learning to balance a bicycle, where humans progress from complete incompetence to near perfection after crashing once or a few times. Clearly, such rapid acquisition of new associations or of new motor skills can confer enormous survival advantages.

In laboratory studies, one prominent instance of one-shot learning is the Bruce effect (*Bruce, 1959*). Here, the female mouse forms an olfactory memory of her mating partner that allows her to terminate the pregnancy if she encounters another male that threatens infanticide. Another form of rapid learning accessible to laboratory experiments is fear conditioning, where a formerly innocuous stimulus gets associated with a painful experience, leading to subsequent avoidance of the stimulus (*Fanselow and Bolles, 1979*; *Bourtchuladze et al., 1994*). These learning systems appear designed for special purposes, they perform very specific associations, and govern binary behavioral decisions. They are likely implemented by specialized brain circuits, and indeed great progress has been made in localizing these operations to the accessory olfactory bulb (*Brennan and Keverne, 1997*) and the cortical amygdala (*LeDoux, 2000*).

In the attempt to identify more generalizable mechanisms of learning and decision making, one route has been to train laboratory animals on abstract tasks with tightly specified sensory inputs that are linked to motor outputs via arbitrary contingency rules. Canonical examples are a monkey reporting motion in a visual stimulus by saccading its eyes (*Newsome and Paré, 1988*), and a mouse in a box classifying stimuli by moving its forelimbs or the tongue (*Burgess et al., 2017*; *Guo et al., 2014*). The tasks are of low complexity, typically a one bit decision based on 1 or 2 bits of input. Remarkably, they are learned exceedingly slowly: a mouse typically requires many weeks of shaping and thousands of trials to reach asymptotic performance; a monkey may require many months (*Carandini and Churchland, 2013*).

\*For correspondence:
perona@caltech.edu (PP);
meister@caltech.edu (MM)

[†]These authors contributed
equally to this work

Competing interest: See
page 27

Reviewing editor: Mackenzie W
Mathis, EPFL, Switzerland

What is needed therefore is a rodent behavior that involves complex decision making, with many input variables and many possible choices. Ideally, the animals would learn to perform this task without excessive intervention by human shaping, so we may be confident that they employ innate brain mechanisms rather than circuits created by the training. Obviously, the behavior should be easy to measure in the laboratory. Finally, it would be satisfying if this behavior showed a glimpse of rapid learning.

Navigation through space is a complex behavior displayed by many animals. It typically involves integrating multiple cues to decide among many possible actions. It relies intimately on rapid learning. For example, a pigeon or desert ant leaving its shelter acquires the information needed for the homing path in a single episode. Major questions remain about how the brain stores this information and converts it to a policy for decisions during the homing path. One way to formalize the act of decision-making in the laboratory is to introduce structure in the environment in the form of a maze that defines straight paths and decision points. A maze of tunnels is in fact a natural environment for a burrowing rodent. Early studies of rodent behavior did place the animals into true labyrinths (*Small, 1901*), but their use gradually declined in favor of linear tracks or boxes with a single choice point.

We report here on the behavior of laboratory mice in a complex labyrinth of tunnels. A single mouse is placed in a home cage from which it has free access to the maze for one night. No handling, shaping, or training by the investigators is involved. By continuous video-recording and automated tracking, we observe the animal's entire life experience within the labyrinth. Some of the mice are water-deprived and a single location deep inside the maze offers water. We find that these animals learn to navigate to the water port after just a few reward experiences. In many cases, one can identify unique moments of 'insight' when the animal's behavior changes discontinuously. This all happens within ~1 h. Underlying the rapid learning is an efficient mode of exploration driven by simple navigation rules. Mice that do not lack water show the same patterns of exploration. This laboratory-based navigation behavior may form a suitable substrate for studying the neural mechanisms that implement few-shot learning.

## Results

### Adaptation to the maze

At the start of the experiment, a single mouse was placed in a conventional mouse cage with bedding and food. A short tunnel offered free access to a maze consisting of a warren of corridors (*Figure 1A–B*). The bottom and walls of the maze were constructed of black plastic that is transparent in the infrared. A video camera placed below the maze captured the animal's actions continuously using infrared illumination (*Figure 1B*). The recordings were analyzed offline to track the movements of the mouse, with keypoints on the nose, mid-body, tail base, and the four feet (*Figure 1D*). All observations were made in darkness during the animal's subjective night.

The logical structure of the maze is a binary tree, with 6 levels of branches, leading from the single entrance to 64 endpoints (*Figure 1C*). A total of 63 T-junctions are connected by straight corridors in a design with maximal symmetry (*Figure 1A*, *Figure 3—figure supplement 1*), such that all the nodes at a given level of the tree have the same local geometry. One of the 64 endpoints of the maze is outfitted with a water port. After activation by a brief nose poke, the port delivers a small drop of water, followed by a 90 s time-out period.

After an initial period of exploratory experiments, we settled on a frozen protocol that was applied to 20 animals. Ten of these mice had been mildly water-deprived for up to 24 h; they received food in the home cage and water only from the port hidden in the maze. Another ten mice had free access to food and water in the cage, and received no water from the port in the maze. Each animal's behavior in the maze was recorded continuously for 7 h during the first night of its experience with the maze, starting the moment the connection tunnel was opened (sample videos here). The investigator played no role during this period, and the animal was free to act as it wished including travel between the cage and the maze.

All the mice except one passed between the cage and the maze readily and frequently (*Figure 1—figure supplement 1*). The single outlier animal barely entered the maze and never progressed past the first junction; we excluded this mouse's data from subsequent analysis. On average

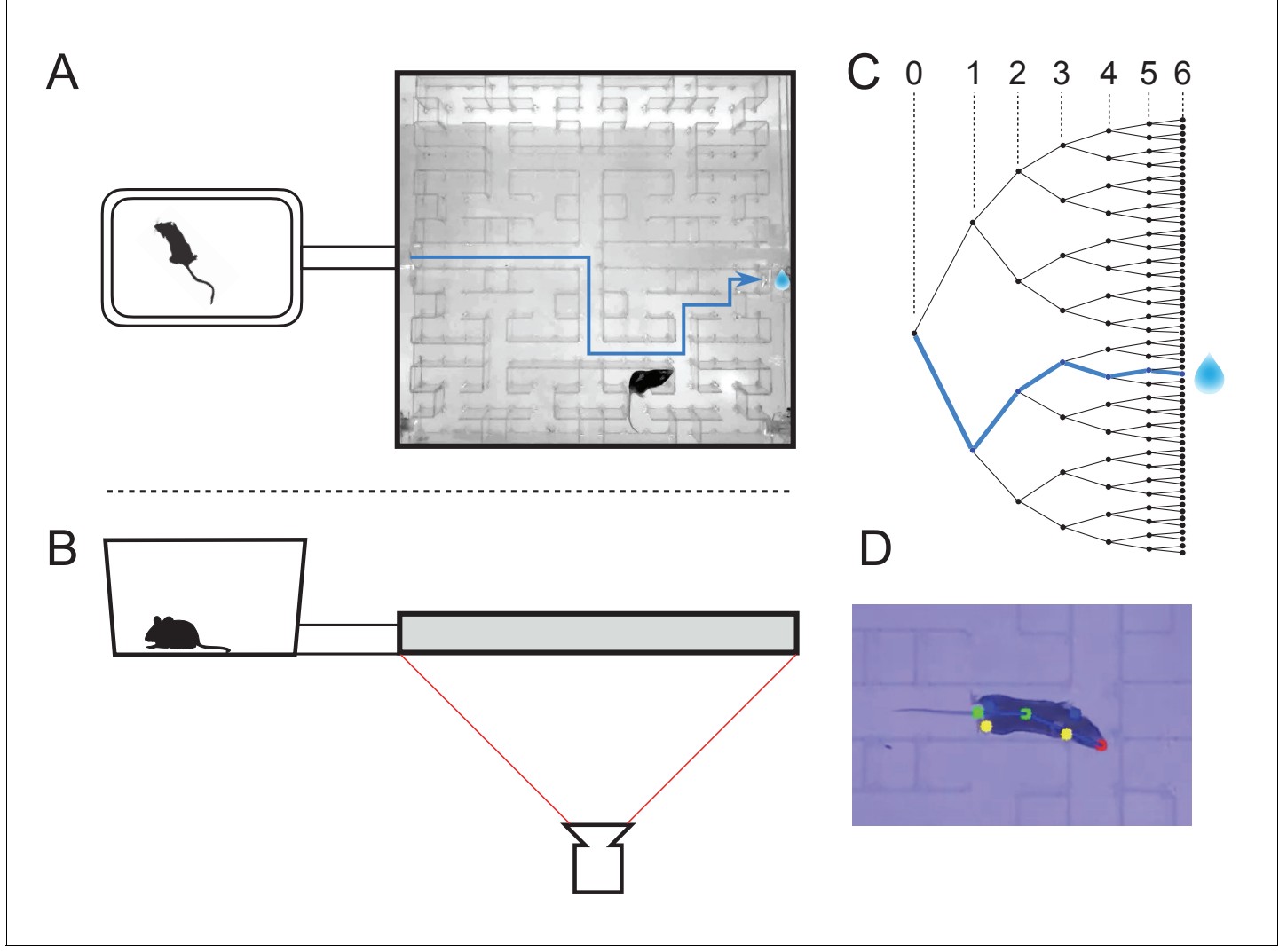

**Figure 1.** The maze environment. Top (**A**) and side (**B**) views of a home cage, connected via an entry tunnel to an enclosed labyrinth. The animal's actions in the maze are recorded via video from below using infrared illumination. (**C**) The maze is structured as a binary tree with 63 branch points (in levels numbered 0,. . .,5) and 64 end nodes. One end node has a water port that dispenses a drop when it gets poked. Blue line in A and C: path from maze entry to water port. (**D**) A mouse considering the options at the maze's central intersection. Colored keypoints are tracked by DeepLabCut: nose, mid body, tail base, four feet.

The online version of this article includes the following figure supplement(s) for figure 1:

**Figure supplement 1.** Occupancy of the maze.

**Figure supplement 2.** Fraction of time in maze by group.

**Figure supplement 3.** Transitions between cage and maze.

over the entire period of study the animals spent 46% of the time in the maze (*Figure 1—figure supplement 2*). This fraction was similar whether or not the animal was motivated by water rewards (47% for rewarded vs 44% for unrewarded animals). Over time the animals appeared increasingly comfortable in the maze, taking breaks for grooming and the occasional nap. When the investigator lifted the cage lid at the end of the night some animals were seen to escape into the safety of the maze.

We examined the rate of transitions from the cage to the maze and how it depends on time spent in the cage (*Figure 1—figure supplement 3A*). Surprisingly the rate of entry into the maze is highest immediately after the animal returns to the cage. Then it declines gradually by a factor of 4 over the first minute in the cage and remains steady thereafter. This is a large effect, observed for every individual animal in both the rewarded and unrewarded groups. By contrast the opposite transition,

namely exit from the maze, occurs at an essentially constant rate throughout the visit (*Figure 1—figure supplement 3B*).

The nature of the animal's forays into the maze changed over time. We call each foray from entrance to exit a 'bout'. After a few hesitant entries into the main corridor, the mouse engaged in one or more long bouts that dove deep into the binary tree to most or all of the leaf nodes (*Figure 2A*). For a water-deprived animal, this typically led to discovery of the reward port. After ~10 bouts, the trajectories became more focused, involving travel to the reward port and some additional exploration (*Figure 2B*). At a later stage still, the animal often executed perfect exploitation bouts that led straight to the reward port and back with no wrong turns (*Figure 2C*). Even at this late stage, however, the animal continued to explore other parts of the maze (*Figure 2D*). Similarly the unrewarded animals explored the maze throughout the night (*Figure 1—figure supplement 2*).

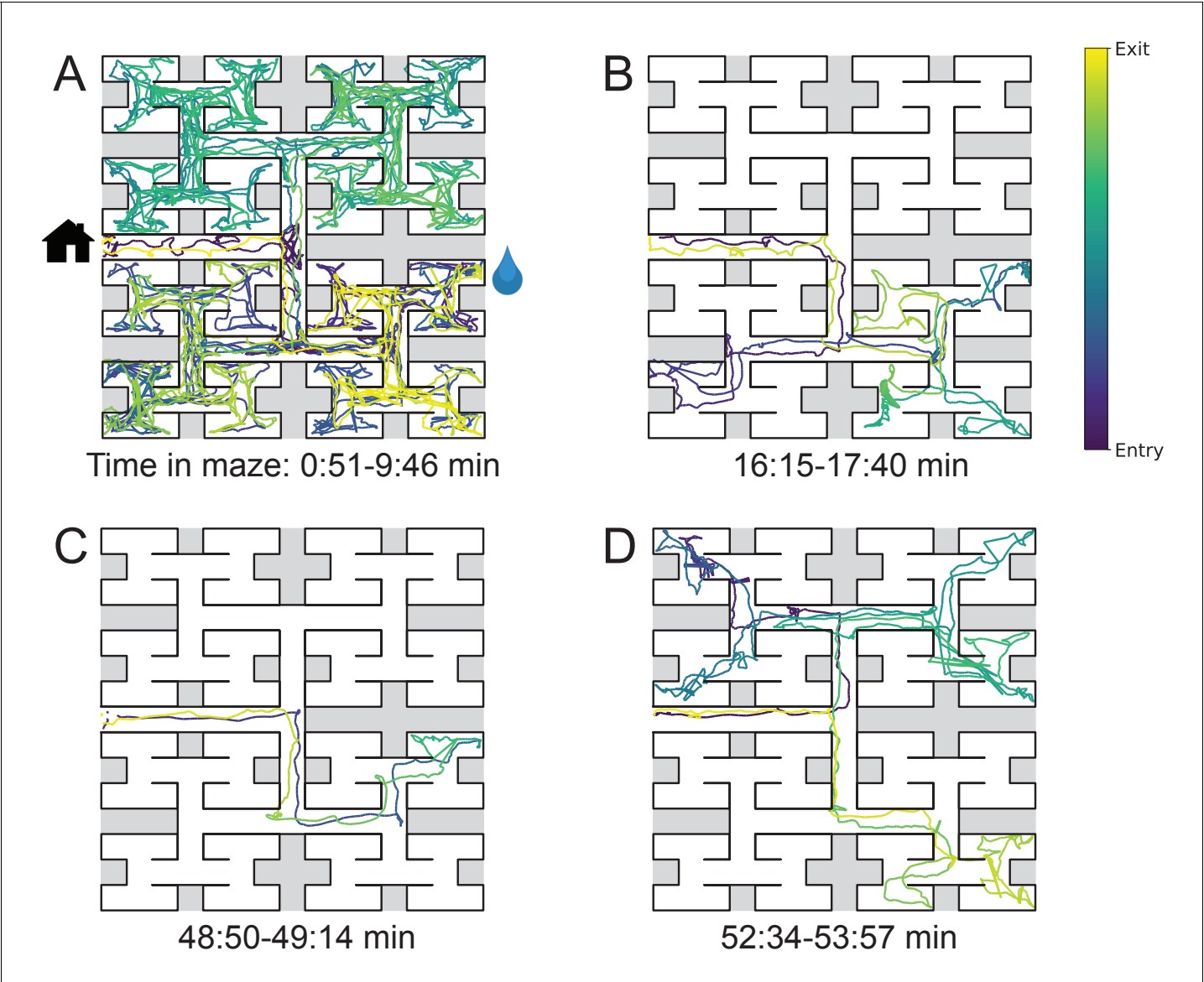

**Figure 2.** Sample trajectories during adaptation to the maze. Four sample bouts from one mouse (B3) into the maze at various times during the experiment (time markings at bottom). The trajectory of the animal's nose is shown; time is encoded by the color of the trace. The entrance from the home cage and the water port are indicated in panel A.

The online version of this article includes the following figure supplement(s) for figure 2:

**Figure supplement 1.** Speed of locomotion.

While the length and structure of the animal's trajectories changed over time, the speed remained remarkably constant after ~50 s of adaptation (*Figure 2—figure supplement 1*).

While *Figure 2* illustrates the trajectory of a mouse's nose in full spatio-temporal detail, a convenient reduced representation is the 'node sequence'. This simply marks the events when the animal enters each of the 127 nodes of the binary tree that describes the maze (see Materials and methods and *Figure 3—figure supplement 1*). Among these nodes, 63 are T-junctions where the animal has three choices for the next node, and 64 are end nodes where the animal's only choice is to reverse course. We call the transition from one node to the next a 'step'. The analysis in the rest of the paper was carried out on the animal's node sequence.

## Few-shot learning of a reward location

We now examine early changes in the animal's behavior when it rapidly acquires and remembers information needed for navigation. First, we focus on navigation to the water port.

The 10 water-deprived animals had no indication that water would be found in the maze. Yet, all 10 discovered the water port in less than 2000 s and fewer than 17 bouts (*Figure 3A*). The port dispensed only a drop of water followed by a 90 s timeout before rearming. During the timeout, the animals generally left the port location to explore other parts of the maze or return home, even though they were not obliged to do so. For each of the water-deprived animals, the frequency at which it consumed rewards in the maze increased rapidly as it learned how to find the water port, then settled after a few reward experiences (*Figure 3A*).

How many reward experiences are sufficient to teach the animal reliable navigation to the water port? To establish a learning curve one wants to compare performance on the identical task over successive trials. Recall that this experiment has no imposed trial structure. Yet the animals naturally segmented their behavior through discrete visits to the maze. Thus, we focused on all the instances when the animal started at the maze entrance and walked to the water port (*Figure 3B*).

On the first few occasions these paths to water can involve hundreds of steps between nodes and their length scatters over a wide range. However, after a few rewards, the animals began taking the perfect path without detours (six steps, *Figure 3—figure supplement 1*), and soon that became the norm. Note the path length plotted here is directly related to the number of 'turning errors': every time the mouse turns away from the shortest path to the water port that adds two steps to the path length (*Equation 7*). The rate of these errors declined over time, by a factor of $e$ after ~10 rewards consumed (*Figure 3B*). Late in the night ~75% of the paths to water were perfect. The animals executed them with increasing speed; eventually, these fast 'water runs' took as little as 2 s (*Figure 3B*). Many of these visits went unrewarded owing to the 90 s timeout period on the water port.

In summary, after ~10 reward experiences on average the mice learn to navigate efficiently to the water port, which requires making six correct decisions, each among three options. Note that even at late times, long after they have perfected the 'water run', the animals continue to take some extremely long paths: a subject for a later section (Figure 7).

## The role of cues attached to the maze

These observations of rapid learning raise the question 'How do the animals navigate?' In particular, does the mouse build an internal representation that guides its action at every junction? Or does it place marks in the external environment that signal the route to the water port? In an extreme version of externalized cognition, the mouse leaves behind a trail of urine marks or other secretions as it walks away from the water port, and on a subsequent bout simply sniffs its way up the odor gradient (*Figure 4A*). This would require no internal representation.

The following experiment offers some partial insights. Owing to the design of the labyrinth one can rotate the entire apparatus by 180 degrees, open one wall and close another, and obtain a maze with the same structure (*Figure 4A*). Alternatively one can also rotate only the floor. After such a modification, all the physical cues attached to the rotated parts now point in the wrong direction, namely to the end node 180 degrees opposite the water port (the 'image location'). If the animal navigated to the goal following cues previously deposited in the maze, it should end up at that image location.

We performed a maze rotation on four animals after several hours of exposure, when they had acquired the perfect route to water. Immediately after rotation, three of the four animals went to

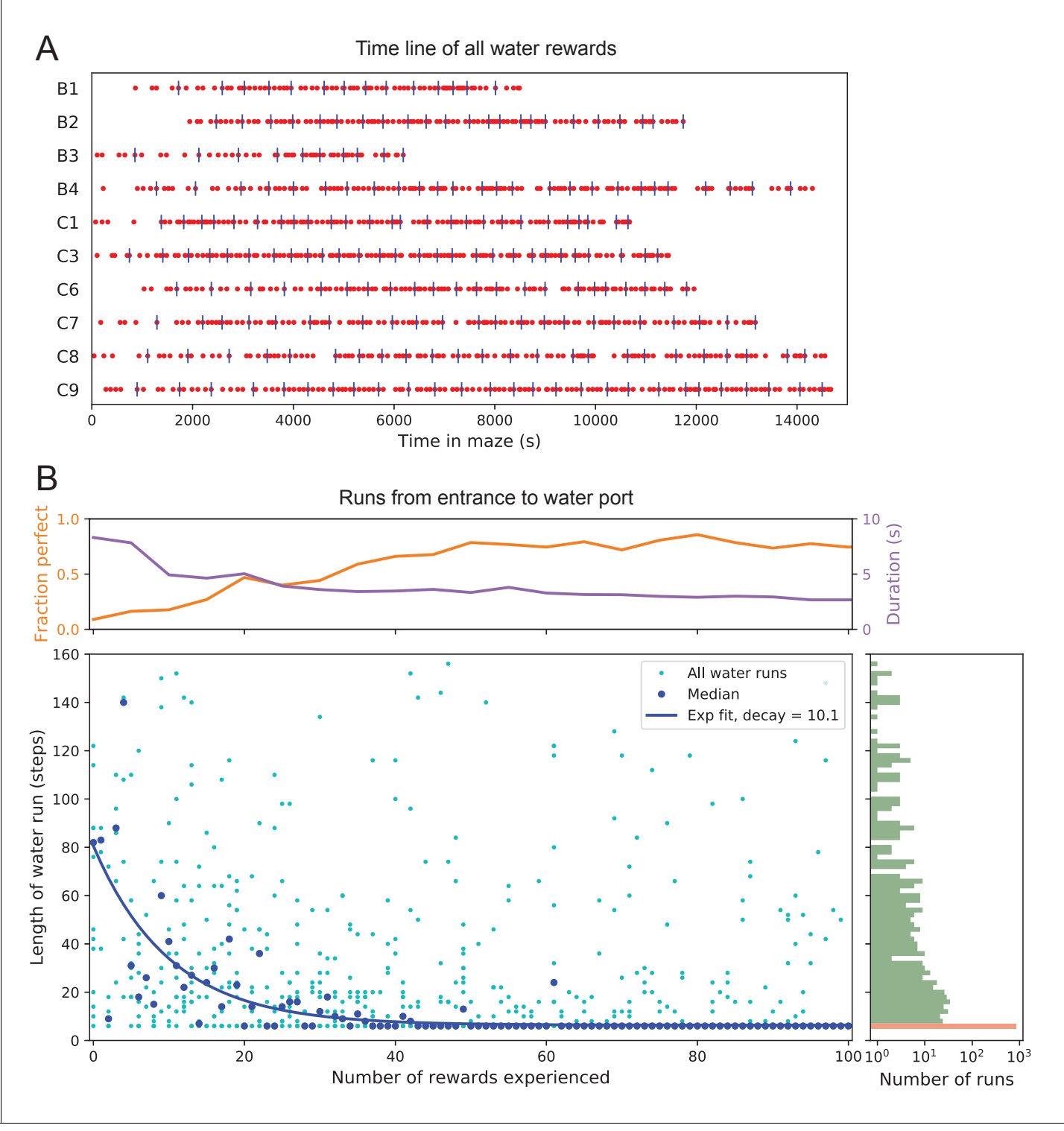

**Figure 3.** Few-shot learning of path to water. (**A**) Time line of all water rewards collected by 10 water-deprived mice (red dots, every fifth reward has a blue tick mark). (**B**) The length of runs from the entrance to the water port, measured in steps between nodes, and plotted against the number of rewards experienced. Main panel: All individual runs (cyan dots) and median over 10 mice (blue circles). Exponential fit decays by $1/e$ over 10.1 rewards. Right panel: Histogram of the run length, note log axis. Red: perfect runs with the minimum length 6; green: longer runs. Top panel: The fraction of perfect runs (length 6) plotted against the number of rewards experienced, along with the median duration of those perfect runs.

The online version of this article includes the following figure supplement(s) for figure 3:

**Figure supplement 1.** Definition of node trajectories.

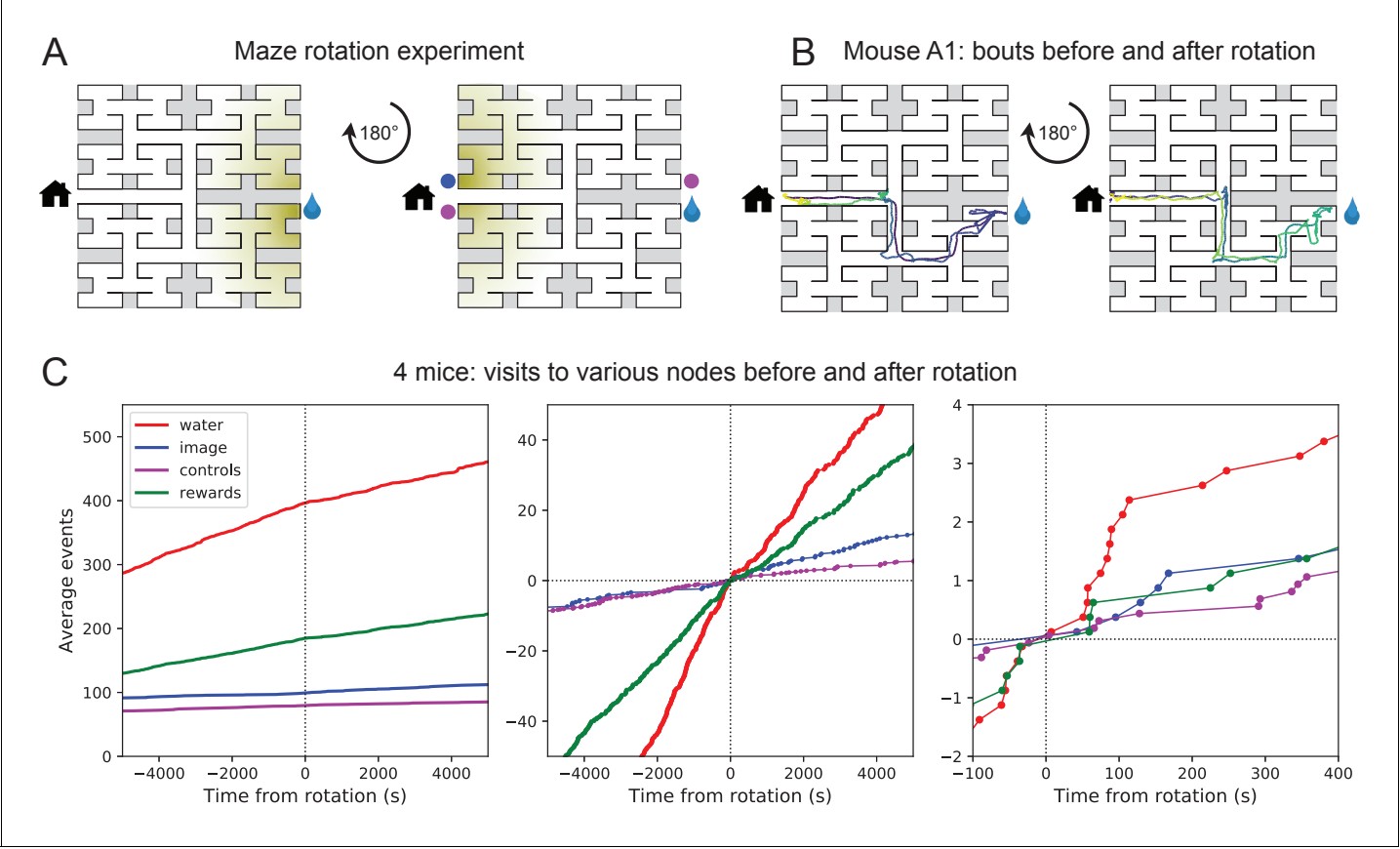

**Figure 4.** Navigation is robust to rotation of the maze. (**A**) Logic of the experiment: The animal may have deposited an odorant in the maze (shading) that is centered on the water port. After 180 degree rotation of the maze, that gradient would lead to the image of the water port (blue dot). We also measure how often the mouse goes to two control nodes (magenta dots) that are related by symmetry. (**B**) Trajectory of mouse 'A1' in the bouts immediately before and after maze rotation. Time coded by color from dark to light as in *Figure 2*. (**C**) Left: Cumulative number of rewards as well as visits to the water port, the image of the water port, and the control nodes. All events are plotted vs time before and after the maze rotation. Average over four animals. Middle and right: Same data with the counts centered on zero and zoomed in for better resolution.

The online version of this article includes the following figure supplement(s) for figure 4:

**Figure supplement 1.** Navigation before and after maze rotation for each animal.

**Figure supplement 2.** Speed before and after maze rotation.

the correct water port on their first entry into the maze, and before ever visiting the image location (e.g. *Figure 4B*). The fourth mouse visited the image location once and then the correct water port (*Figure 4—figure supplement 1*). The mice continued to collect water rewards efficiently even immediately after the rotation.

Nonetheless, the maze rotation did introduce subtle changes in behavior that lasted for an hour or more (*Figure 4C*). Visits to the image location were at chance levels prior to rotation, then increased by a factor of 1.8. Visits to the water port declined in frequency, although they still exceeded visits to the image location by a factor of 5. The reward rate declined by a factor of 0.7. These effects could be verified for each animal (*Figure 4—figure supplement 1*). The speed of the mice was not disturbed (*Figure 4—figure supplement 2*).

In summary, for navigation to the water port the experienced animals do not strictly depend on physical cues that are attached to the maze. This includes any material they might have deposited, but also pre-existing construction details by which they may have learned to identify locations in the maze. The mice clearly notice a change in these cues, but continue to navigate effectively to the goal. This conclusion applies to the time point of the rotation, a few hours into the experiment. Conceivably, the animal's navigation policy and its use of sensory cues changes in the course of learning.

This and many other questions regarding the mechanisms of cognition will be taken up in a separate study.

## Discontinuous learning

While an average across animals shows evidence of rapid learning (*Figure 3*) one wonders whether the knowledge is acquired gradually or discontinuously, through moments of 'sudden insight'. To explore this we scrutinized more closely the time line of individual water-deprived animals in their experience with the maze. The discovery of the water port and the subsequent collection of water drops at a regular rate is one clear change in behavior that relies on new knowledge. Indeed, the rate of water rewards can increase rather suddenly (*Figure 3A*), suggesting an instantaneous step in knowledge.

Over time, the animals learned the path to water not only from the entrance of the maze but from many locations scattered throughout the maze. The largest distance between the water port and an end node in the opposite half of the maze involves 12 steps through 11 intersections (*Figure 5A*). Thus, we included as another behavioral variable the occurrence of long direct paths to the water port which reflects how directly the animals navigate within the maze.

*Figure 5B* shows for one animal the cumulative occurrence of water rewards and that of long direct paths to water. The animal discovers the water port early on at 75 s, but at 1380 s the rate of water rewards jumps suddenly by a factor of 5. The long paths to water follow a rather different time line. At first they occur randomly, at the same rate as the paths to the unrewarded control nodes. At 2070 s the long paths suddenly increase in frequency by a factor of 5. Given the sudden change in rates of both kinds of events there is little ambiguity about when the two steps happen and they are well separated in time (*Figure 5B*).

The animal behaves as though it gains a new insight at the time of the second step that allows it to travel to the water port directly from elsewhere in the maze. Note that the two behavioral variables are independent: The long paths don't change when the reward rate steps up, and the reward rate doesn't change when the rate of long paths steps up. Another animal (*Figure 5C*) similarly showed an early step in the reward rate (at 860 s) and a dramatic step in the rate of long paths (at 2580 s). In this case, the emergence of long paths coincided with a modest increase (factor of 2) in the reward rate.

Similar discontinuities in behavior were seen in at least 5 of the 10 water-deprived animals (*Figure 5B*, *Figure 5—figure supplement 1*, *Figure 5—source data 1*), and their timing could be identified to a precision of ~200 s. More gradual performance change was observed for the remaining animals (*Figure 5D*). We varied the criterion of performance by asking for even longer error-free paths, and the results were largely unchanged and no additional discontinuity appeared. These observations suggest that mice can acquire a complex decision-making skill rather suddenly. A mouse may have multiple moments of sudden insight that affect different aspects of its behavior. The exact time of the insight cannot be predicted but is easily identified post-hoc. Future neurophysiological studies of the phenomenon will face the interesting challenge of capturing these singular events.

## One-shot learning of the home path

For an animal entering an unfamiliar environment, the most important path to keep in memory may be the escape route. In the present case that is the route to the maze entrance, from which the tunnel leads home to the cage. We expected that the mice would begin by penetrating into the maze gradually and return home repeatedly so as to confirm the escape route, a pattern previously observed for rodents in an open arena (*Tchernichovski et al., 1998*; *Fonio et al., 2009*). This might help build a memory of the home path gradually level-by-level into the binary tree. Nothing could be further from the truth.

At the end of any given bout into the maze, there is a 'home run', namely the direct path without reversals that takes the animal to the exit (see *Figure 3—figure supplement 1*). *Figure 6A* shows the nodes where each animal started its first home run, following the first penetration into the maze. With few exceptions, that first home run began from an end node, as deep into the maze as possible. Recall that this involves making the correct choice at six successive three-way intersections, an outcome that is unlikely to happen by chance.

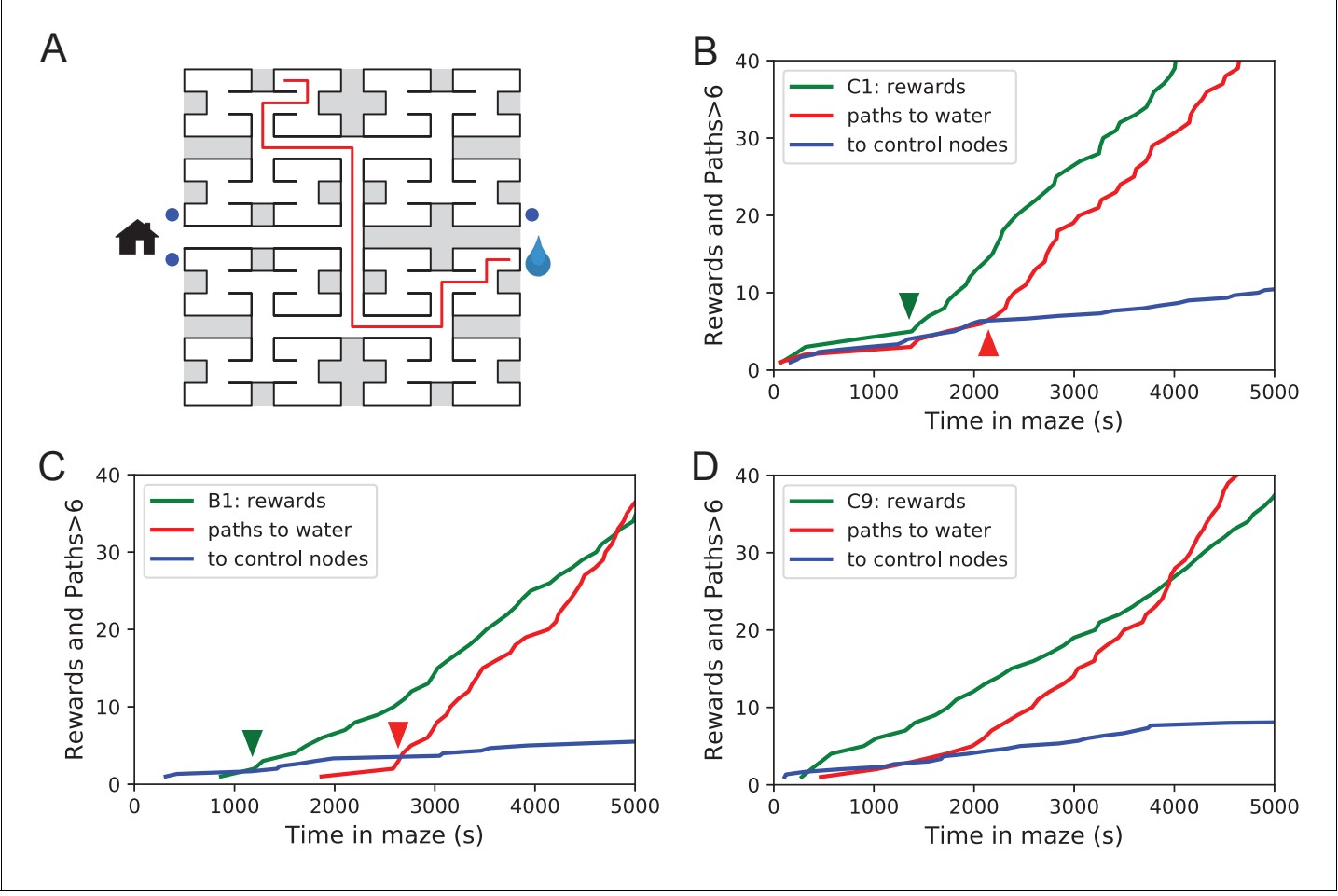

**Figure 5.** Sudden changes in behavior. (**A**) An example of a long uninterrupted path through 11 junctions to the water port (drop icon). Blue circles mark control nodes related by symmetry to the water port to assess the frequency of long paths occurring by chance. (**B**) For one animal (named C1) the cumulative number of rewards (green); of long paths (>6 junctions) to the water port (red); and of similar paths to the three control nodes (blue, divided by 3). All are plotted against the time spent in the maze. Arrowheads indicate the time of sudden changes, obtained from fitting a step function to the rates. (**C**) Same as B for animal B1. (**D**) Same as B for animal C9, an example of more continuous learning.

The online version of this article includes the following source data and figure supplement(s) for figure 5:

**Source data 1.** Statistics of sudden changes in behavior.

**Figure supplement 1.** Long direct paths for all animals.

The above hypothesis regarding gradual practice of home runs would predict that short home runs should appear before long ones in the course of the experiment. The opposite is the case (*Figure 6B*). In fact, the end nodes (level 7 of the maze) are by far the favorite place from which to return to the exit, and those maximal-length home runs systematically appear before shorter ones. This conclusion was confirmed for each individual animal, whether rewarded or unrewarded.

Clearly, the animals do not practice the home path or build it up gradually. Instead they seem to possess an Ariadne's thread (*Apollodorus, 1921*) starting with their first excursion into the maze, long before they might have acquired any general knowledge of the maze layout. On the other hand, the mouse does not follow the strategy of Theseus, namely to precisely retrace the path that led it into the labyrinth. In that case the animal's home path should be the reverse of the path into the maze that started the bout. Instead the entry path and the home path tend to have little overlap (*Figure 6C*). Note the minimum overlap is 1, because all paths into and out of the maze have to pass through the central junction (node 0 in *Figure 3—figure supplement 1*). This is also the most frequent overlap. The peak at overlaps 6–8 for rewarded animals results from the frequent paths to the water port and back, a sequence of at least seven nodes in each direction. The separation of

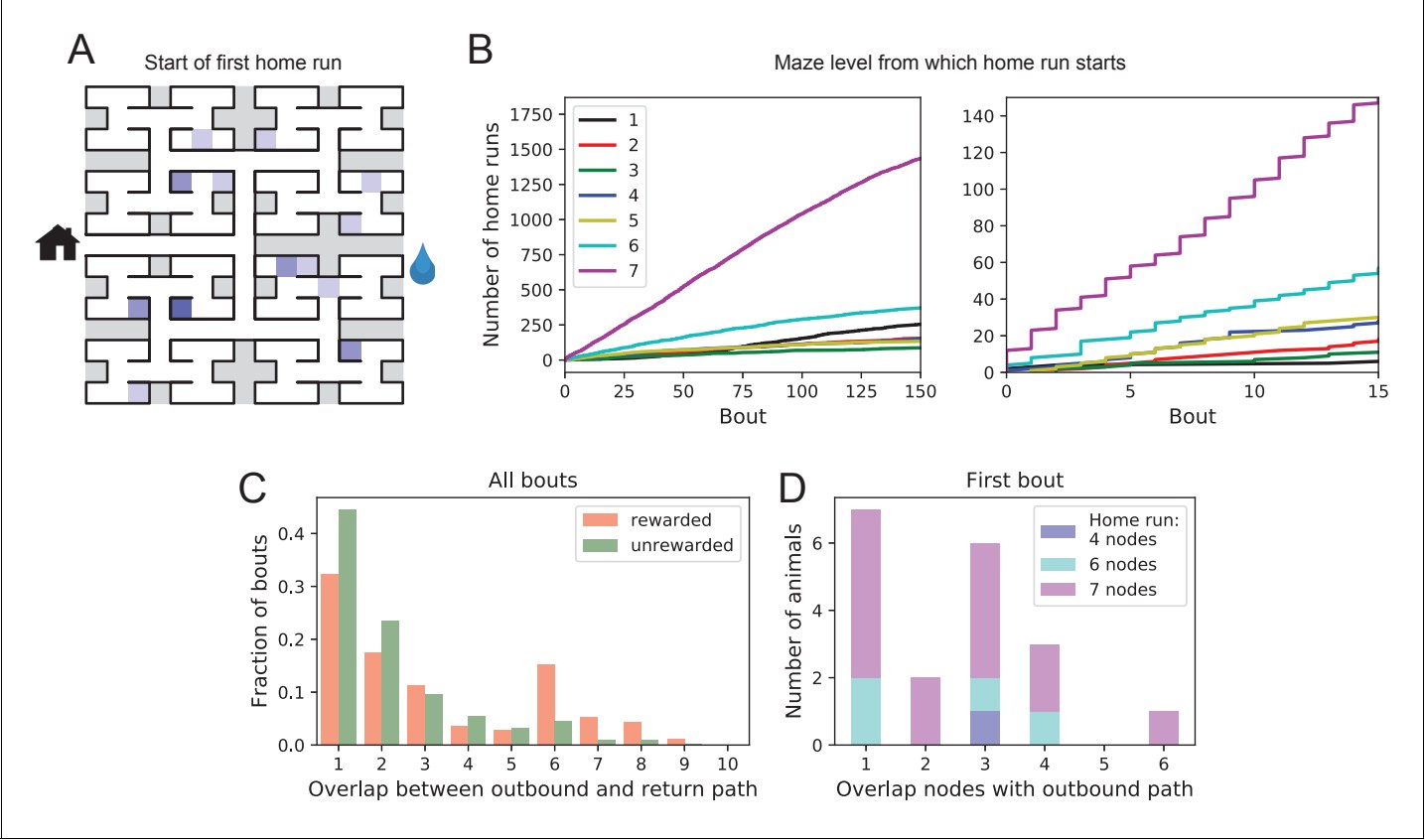

**Figure 6.** Homing succeeds on first attempt. (**A**) Locations in the maze where the 19 animals started their first return to the exit (home run). Some locations were used by two or three animals (darker color). (**B**) Left: The cumulative number of home runs from different levels in the maze, summed over all animals, and plotted against the bout number. Level 1 = first T-junction, level 7 = end nodes. Right: Zoom of (Left) into early bouts. (**C**) Overlap between the outbound and the home path. Histogram of the overlap for all bouts of all animals. (**D**) Same analysis for just the first bout of each animal. The length of the home run is color-coded as in panel B.

outbound and return path is seen even on the very first home run (*Figure 6D*). Many home runs from the deepest level (seven nodes) have only the central junction in common with the outbound path (overlap = 1).

In summary, it appears that the animal acquires a homing strategy over the course of a single bout, and in a manner that allows a direct return home even from locations not previously encountered.

## Structure of behavior in the maze

Here, we focus on rules and patterns that govern the animal's activity in the maze on both large and small scales.

### Behavioral states

Once the animal has learned to perform long uninterrupted paths to the water port, one can categorize its behavior within the maze by three states: (1) walking to the water port; (2) walking to the exit; and (3) exploring the maze. Operationally we define exploration as all periods in which the animal is in the maze but not on a direct path to water or to the exit. For the 10 sated animals this includes all times in the maze except for the walks to the exit.

*Figure 7* illustrates the occupancies and transition probabilities between these states. The animals spent most of their time by far in the exploration state: 84% for rewarded and 95% for unrewarded mice. Across animals there was very little variation in the balance of the three modes (*Figure 7— source data 1*). The rewarded mice began about half their bouts into the maze with a trip to the

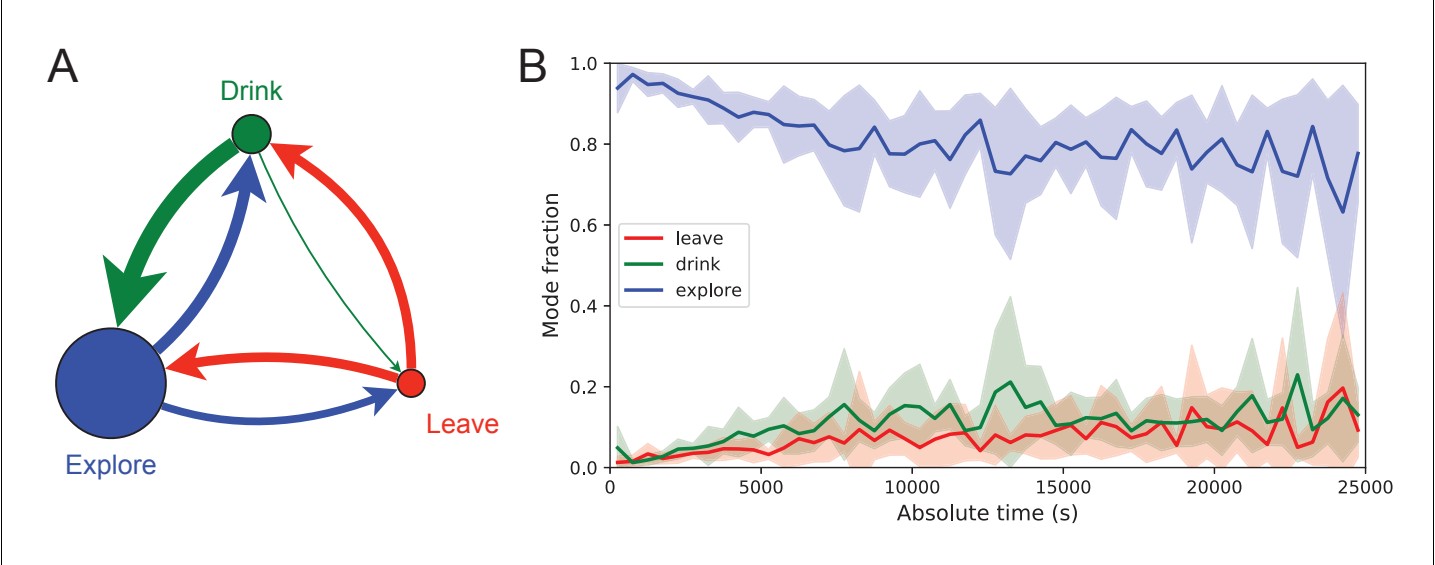

**Figure 7.** Exploration is a dominant and persistent mode of behavior. (**A**) Ethogram for rewarded animals. Area of the circle reflects the fraction of time spent in each behavioral mode averaged over animals and duration of the experiment. Width of the arrow reflects the probability of transitioning to another mode. 'Drink' involves travel to the water port and time spent there. Transitions from 'Leave' represent what the animal does at the start of the next bout into the maze. (**B**) The fraction of time spent in each mode as a function of absolute time throughout the night. Mean ± SD across the 10 rewarded animals.

The online version of this article includes the following source data for figure 7:

**Source data 1.** Three modes of behavior.

water port and the other half by exploring (*Figure 7A*). After a drink, the animals routinely continued exploring, about 90% of the time.

For water-deprived animals, the dominance of exploration persisted even at a late stage of the night when they routinely executed perfect exploitation bouts to and from the water port: Over the duration of the night the 'explore' fraction dropped slightly from 0.92 to 0.75, with the balance accrued to the 'drink' and 'leave' modes as the animals executed many direct runs to the water port and back. The unrewarded group of animals also explored the maze throughout the night even though it offered no overt rewards (*Figure 7—source data 1*). One suspects that the animals derive some intrinsic reward from the act of patrolling the environment itself.

## Efficiency of exploration

During the direct paths to water and to the exit the animal behaves deterministically, whereas the exploration behavior appears stochastic. Here, we delve into the rules that govern the exploration component of behavior.

One can presume that a goal of the exploratory mode is to rapidly survey all parts of the environment for the appearance of new resources or threats. We will measure the efficiency of exploration by how rapidly the animal visits all end nodes of the binary maze, starting at any time during the experiment. The optimal agent with perfect memory and complete knowledge of the maze – including the absence of any loops – could visit the end nodes systematically one after another without repeats, thus encountering all of them after just 64 visits. A less perfect agent, on the other hand, will visit the same node repeatedly before having encountered all of them. *Figure 8A* plots for one exploring mouse the number of distinct end nodes it encountered as a function of the number of end nodes visited. The number of new nodes rises monotonically; 32 of the end nodes have been discovered after the mouse checked 76 times; then the curve gradually asymptotes to 64. We will characterize the efficiency of the search by the number of visits $N_{32}$ required to survey half the end nodes, and define

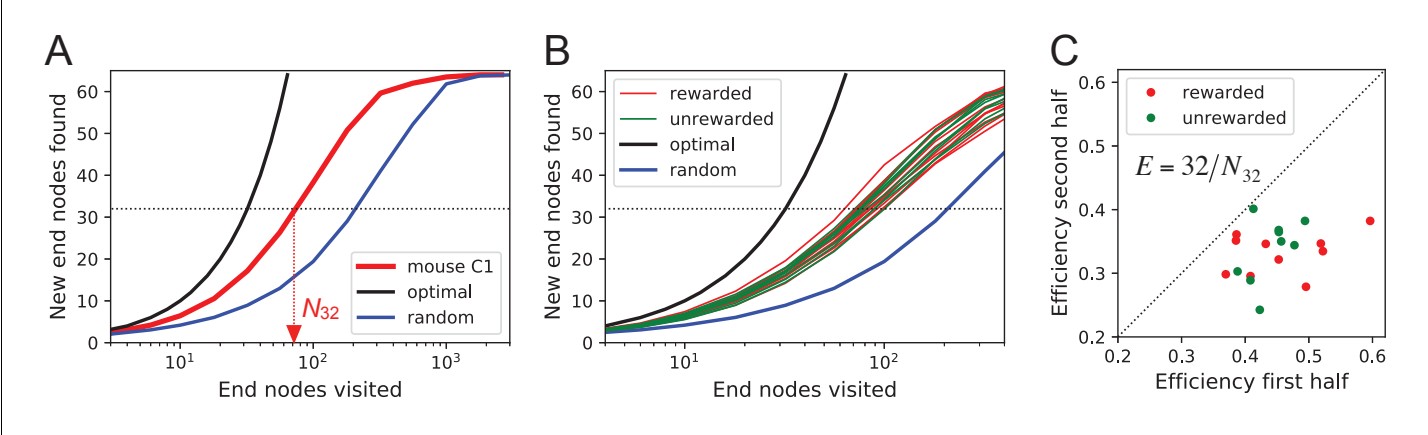

**Figure 8.** Exploration covers the maze efficiently. (A) The number of distinct end nodes encountered as a function of the number of end nodes visited for: mouse C1 (red); the optimal explorer agent (black); an unbiased random walk (blue). Arrowhead: the value $N_{32} = 76$ by which mouse C1 discovered half of the end nodes. (B) An expanded section of the graph in A including curves from 10 rewarded (red) and nine unrewarded (green) animals. The efficiency of exploration, defined as $E = 32/N_{32}$, is $0.385 \pm 0.050$ (SD) for rewarded and $0.384 \pm 0.039$ (SD) for unrewarded mice. (C) The efficiency of exploration for the same animals, comparing the values in the first and second halves of the time in the maze. The decline is a factor of $0.74 \pm 0.12$ (SD) for rewarded and $0.81 \pm 0.13$ (SD) for unrewarded mice.

The online version of this article includes the following figure supplement(s) for figure 8:

**Figure supplement 1.** Efficiency of exploration.

$$E = \frac{32}{N_{32}} \tag{1}$$

This mouse explores with efficiency $E = 32/76 = 0.42$. For comparison, *Figure 8A* plots the performance of the optimal agent ($E = 1.0$) and that of a random walker that makes random decisions at every three-way junction ($E = 0.23$). Note the mouse is about half as efficient as the optimal agent, but twice as efficient as a random walker.

The different mice were remarkably alike in this component of their exploratory behavior (*Figure 8B*): across animals the efficiency varied by only 11% of the mean ($0.387 \pm 0.044$ SD). Furthermore, there was no detectable difference in efficiency between the rewarded animals and the sated unrewarded animals. Over the course of the night, the efficiency declined significantly for almost every animal – whether rewarded or not – by an average of 23% (*Figure 8C*).

## Rules of exploration

What allows the mice to search much more efficiently than a random walking agent? We inspected more closely the decisions that the animals make at each three-way junction. It emerged that these decisions are governed by strong biases (*Figure 9*). The probability of choosing each arm of a T-junction depends crucially on how the animal entered the junction. The animal can enter a T-junction from three places and exit it in three directions (*Figure 9A*). By tallying the frequency of all these occurrences across all T-junctions in the maze one finds clear deviations from an unbiased random walk (*Figure 9B*, *Figure 9—source data 1*).

First, the animals have a strong preference for proceeding through a junction rather than returning to the preceding node ($P_{SF}$ and $P_{BF}$ in *Figure 9B*). Second, there is a bias in favor of alternating turns left and right rather than repeating the same direction turn ($P_{SA}$). Finally, the mice have a mild preference for taking a branch off the straight corridor rather than proceeding straight ($P_{BS}$). A comparison across animals again revealed a remarkable degree of consistency even in these local rules of behavior: The turning biases varied by only 3% across the population and even between the rewarded and unrewarded groups (*Figure 9B*, *Figure 9—source data 1*).

Qualitatively, one can see that these turning biases will improve the animal's search strategy. The forward biases $P_{SF}$ and $P_{BF}$ keep the animal from re-entering territory it has covered already. The

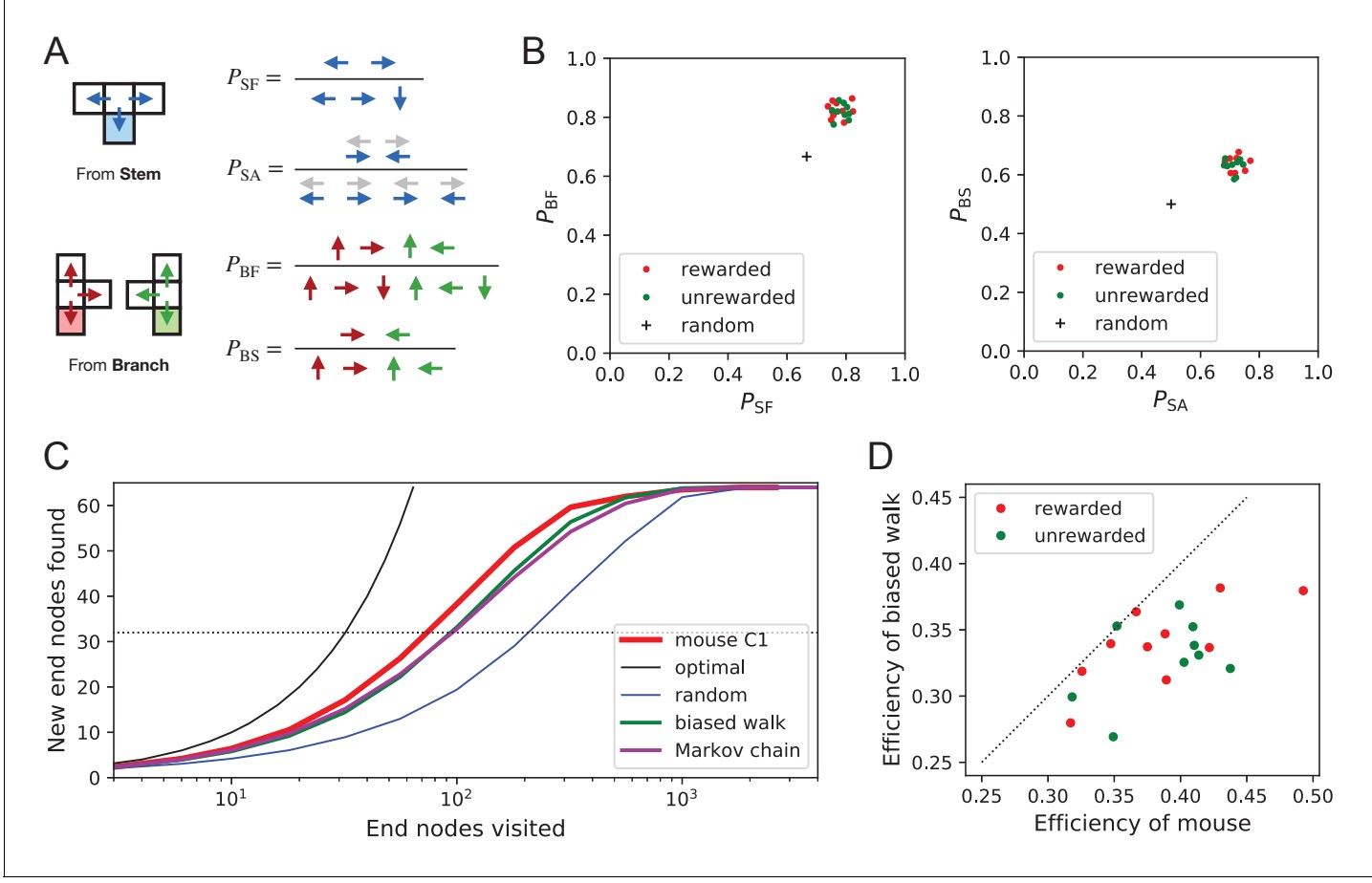

**Figure 9.** Turning biases favor exploration. (**A**) Definition of four turning biases at a T-junction based on the ratios of actions taken. Top: An animal arriving from the stem of the T (shaded) may either reverse or turn left or right. $P_{SF}$ is the probability that it will move forward rather than reversing. Given that it moves forward, $P_{SA}$ is the probability that it will take an alternating turn from the preceding one (gray), that is left-right or right-left. Bottom: An animal arriving from the bar of the T may either reverse or go straight, or turn into the stem of the T. $P_{BF}$ is the probability that it will move forward through the junction rather than reversing. Given that it moves forward, $P_{BS}$ is the probability that it turns into the stem. (**B**) Scatter graph of the biases $P_{SF}$ and $P_{BF}$ (left) and $P_{SA}$ and $P_{BS}$ (right). Every dot represents a mouse. Cross: values for an unbiased random walk. (**C**) Exploration curve of new end nodes discovered vs end nodes visited, displayed as in *Figure 8A*, including results from a biased random walk with the four turning biases derived from the same mouse, as well as a more elaborate Markov-chain model (see *Figure 11C*). (**D**) Efficiency of exploration (*Equation 1*) in 19 mice compared to the efficiency of the corresponding biased random walk.

The online version of this article includes the following source data for figure 9:

**Source data 1.** Bias statistics.

bias $P_{BS}$ favors taking a branch that leads out of the maze. This allows the animal to rapidly cross multiple levels during an outward path and then enter a different territory. By comparison, the unbiased random walk tends to get stuck in the tips of the tree and revisits the same end nodes many times before escaping. To test this intuition, we simulated a biased random agent whose turning probabilities at a T-junction followed the same biases as measured from the animal (*Figure 9C*). These biased agents did in fact search with much higher efficiency than the unbiased random walk. They did not fully explain the behavior of the mice (*Figure 9D*), accounting for ~87% of the animal's efficiency (compared to 60% for the random walk). A more sophisticated model of the animal's behavior - involving many more parameters (Figure 11C) - failed to get any closer to the observed efficiency (*Figure 9C*, *Figure 8—figure supplement 1C*). Clearly some components of efficient search in these mice remain to be understood.

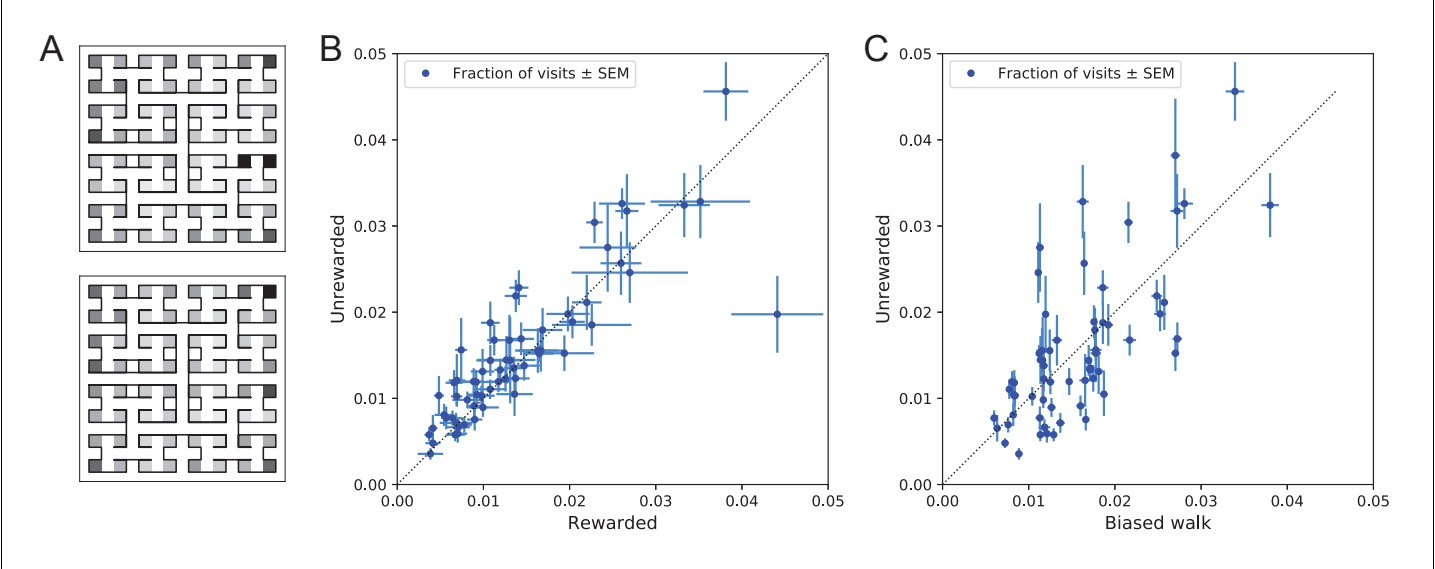

**Figure 10.** Preference for certain end nodes during exploration. (**A**) The number of visits to different end nodes encoded by a gray scale. Top: rewarded, bottom: unrewarded animals. Gray scale spans a factor of 12 (top) or 13 (bottom). (**B**) The fraction of visits to each end node, comparing the rewarded vs unrewarded group of animals. Each data point is for one end node, the error bar is the SEM across animals in the group. The outlier on the bottom right is the neighbor of the water port, a frequently visited end node among rewarded animals. The water port is off scale and not shown. (**C**) As in panel B but comparing the unrewarded animals to their simulated 4-bias random walks. These biases explain 51% of the variance in the observed preference for end nodes.

### Systematic node preferences

A surprising aspect of the animals' explorations is that they visit certain end nodes of the binary tree much more frequently than others (*Figure 10*). This effect is large: more than a factor of 10 difference between the occupancy of the most popular and least popular end nodes (*Figure 10A–B*). This was surprising given our efforts to design the maze symmetrically, such that in principle all end nodes should be equivalent. Furthermore, the node preferences were very consistent across animals and even across the rewarded and unrewarded groups. Note that the standard error across animals of each node's occupancy is much smaller than the differences between the nodes (*Figure 10B*).

The nodes on the periphery of the maze are systematically preferred. Comparing the outermost ring of 26 end nodes (excluding the water port and its neighbor) to the innermost 16 end nodes, the outer ones are favored by a large factor of 2.2. This may relate to earlier reports of a 'centrifugal tendency' among rats patrolling a maze (*Uster et al., 1976*).

Interestingly, the biased random walk using four bias numbers (*Figure 9*, *Figure 11D*) replicates a good amount of the pattern of preferences. For unrewarded animals, where the maze symmetry is not disturbed by the water port, the biased random walk predicts 51% of the observed variance across nodes (*Figure 10C*), and an outer/inner node preference of 1.97, almost matching the observed ratio of 2.20. The more complex Markov-chain model of behavior (*Figure 11C*) performed slightly better, explaining 66% of the variance in port visits and matching the outer/inner node preference of 2.20.

### Models of maze behavior

Moving beyond the efficiency of exploration one may ask more broadly: How well do we really understand what the mouse does in the maze? Can we predict its action at the next junction? Once the predictable component is removed, how much intrinsic randomness remains in the mouse's behavior? Here, we address these questions using more sophisticated models that predict the probability of the mouse's future actions based on the history of its trajectory.

At a formal level, the mouse's trajectory through the maze is a string of numbers standing for the nodes the animal visited (*Figure 11A* and *Figure 3—figure supplement 1*). We want to predict the next action of the mouse, namely the step that takes it to the next node. The quality of the model

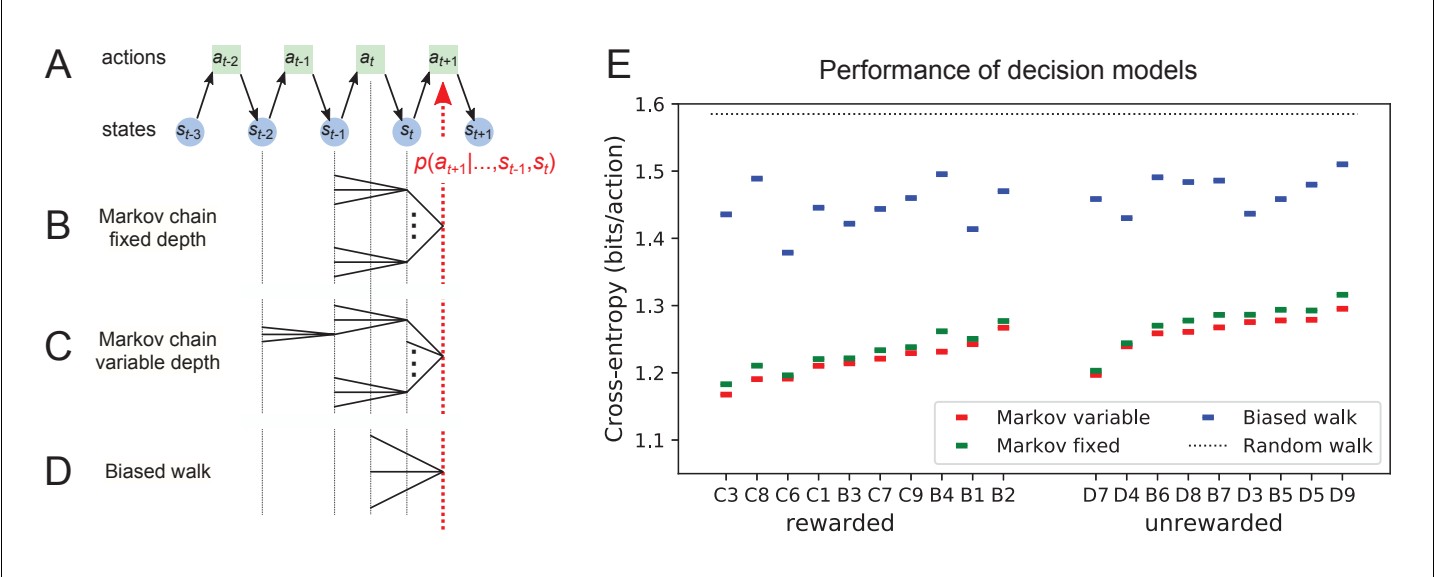

**Figure 11.** Recent history constrains the mouse's decisions. (A) The mouse's trajectory through the maze produces a sequence of states $s_t = \text{node occupied after step } t$. From each state, up to three possible actions lead to the next state (end nodes allow only one action). We want to predict the animal's next action, $a_{t+1}$, based on the prior history of states or actions. (B–D) Three possible models to make such a prediction. (B) A fixed-depth Markov chain where the probability of the next action depends only on the current state $s_t$ and the preceding state $s_{t-1}$. The branches of the tree represent all $3 \times 127$ possible histories $(s_{t-1}, s_t)$. (C) A variable-depth Markov chain where only certain branches of the tree of histories contribute to the action probability. Here one history contains only the current state, some others reach back three steps. (D) A biased random walk model, as defined in *Figure 9*, in which the probability of the next action depends only on the preceding action, not on the state. (E) Performance of the models in (B,C,D) when predicting the decisions of the animal at T-junctions. In each case we show the cross-entropy between the predicted action probability and the real actions of the animal (lower values indicate better prediction, perfect prediction would produce zero). Dotted line represents an unbiased random walk with 1/3 probability of each action.

The online version of this article includes the following figure supplement(s) for figure 11:

**Figure supplement 1.** Markov model fits.

will be assessed by the cross-entropy between the model's predictions and the mouse's observed actions, measured in bits per action. This is the uncertainty that remains about the mouse's next action given the prediction from the model. The ultimate lower limit is the true source entropy of the mouse, namely that component of its decisions that cannot be explained by the history of its actions.

One family of models we considered are fixed-depth Markov chains (*Figure 11B*). Here, the probability of the next action $a_{t+1}$ is specified as a function of the history stretching over the $k$ preceding nodes $(s_{t-k+1}, \ldots, s_t)$. In fitting the model to the mouse's actual node sequence one tallies how often each history leads to each action, and uses those counts to estimate the conditional probabilities $p(a_{t+1}|s_{t-k+1}, \ldots, s_t)$. Given a new node sequence, the model will then use the history strings $(s_{t-k+1}, \ldots, s_t)$ to predict the outcome of the next action. In practice, we trained the model on 80% of the animal's trajectory and tested it by evaluating the cross-entropy on the remaining 20%.

Ideally, the depth $k$ of these action trees would be very large, so as to take as much of the prior history into account as possible. However, one soon runs into a problem of over-fitting: Because each T-junction in the maze has three neighboring junctions, the number of possible histories grows as $3^k$. As $k$ increases, this quickly exceeds the length of the measured node sequence, so that every history appears only zero or one times in the data. At this point, one can no longer estimate any probabilities, and cross-validation on a different segment of data fails catastrophically. In practice, we found that this limitation sets in already beyond $k = 2$ (*Figure 11—figure supplement 1A*). To address this issue of data-limitation, we developed a variable-depth Markov chain (*Figure 11C*). This model retains longer histories, but only if they occur frequently enough to allow a reliable probability estimate (see Materials and methods, *Figure 11—figure supplement 1B–C*). In addition, we

explored different schemes of pooling the counts across certain T-junctions that are related by the symmetry of the maze (see Materials and methods).

With these methods, we focused on the portions of trajectory when the mouse was in 'explore' mode, because the segments in 'drink' and 'leave' mode are fully predictable. Furthermore, we evaluated the models only at nodes corresponding to T-junctions, because the decision from an end node is again fully predictable. *Figure 11E* compares the performance of various models of mouse behavior. The variable-depth Markov chains routinely produced the best fits, although the improvement over fixed-depth models was modest. Across all 19 animals in this study the remaining uncertainty about the animal's action at a T-junction is 1.237 ± 0.035 (SD) bits/action, compared to the prior uncertainty of $log_2$ 3 = 1.585 bits. The rewarded animals have slightly lower entropy than the unrewarded ones (1.216 vs 1.261 bits/action). The Markov chain models that produced the best fits to the behavior used history strings with an average length of ~4.

We also evaluated the predictions obtained from the simple biased random walk model (*Figure 11D*). Recall that this attempts to capture the history-dependence with just four bias parameters (*Figure 9A*). As expected, this produced considerably higher cross-entropies than the more sophisticated Markov chains (by about 18%, *Figure 11E*). Finally, we used several professional file compression routines to try and compress the mouse's node sequence. In principle, this sets an upper bound on the true source entropy of the mouse, even if the compression algorithm has no understanding of animal behavior. The best such algorithm (bzip2 compression *Seward, 2019*) far under-performed all the other models of mouse behavior, giving 43% higher cross-entropy on average, and thus offered no additional useful bounds.

We conclude that during exploration of the maze the mouse's choice behavior is strongly influenced by its current location and ~3 locations preceding it. There are minor contributions from states further back. By knowing the animal's history one can narrow down its action plan at a junction from the a priori 1.59 bits (one of three possible actions) to just ~1.24 bits. This finally is a quantitative answer to the question, 'How well can one predict the animal's behavior?' Whether the remainder represents an irreducible uncertainty – akin to 'free will' of the mouse – remains to be seen. Readers are encouraged to improve on this number by applying their own models of behavior to our published data set.

## Discussion

### Summary of contributions

We present a new approach to the study of learning and decision-making in mice. We give the animal access to a complex labyrinth and leave it undisturbed for a night while monitoring its movements. The result is a rich data set that reveals new aspects of learning and the structure of exploratory behavior. With these methods, we find that mice learn a complex task that requires six correct three-way decisions after only ~10 experiences of success (*Figure 2*, *Figure 3*). Along the way the animal gains task knowledge in discontinuous steps that can be localized to within a few minutes of resolution (*Figure 5*). Underlying the learning process is an exploratory behavior that occupies 90% of the animal's time in the maze and persists long after the task has been mastered, even in complete absence of an extrinsic reward (*Figure 7*). The decisions the animal makes at choice points in the labyrinth are constrained in part by the history of its actions (*Figure 9*, *Figure 11*), in a way that favors efficient searching of the maze (*Figure 8*). This microstructure of behavior is surprisingly consistent across mice, with variation in parameters of only a few percent (*Figure 9*). Our most expressive models to predict the animal's choices still leave a remaining uncertainty of ~1.24 bits per decision (*Figure 11*), a quantitative benchmark by which competing models can be tested. Finally, some of the observations constrain what algorithms the animals might use for learning and navigation (*Figure 4*).

### Historical context

Mazes have been a staple of animal psychology for well over 100 years. The early versions were true labyrinths. For example, *Small, 1901* built a model of the maze in Hampton Court gardens scaled to rat size. Subsequent researchers felt less constrained by Victorian landscapes and began to simplify the maze concept. Most commonly the maze offered one standard path from a starting location to a

food reward box. A few blind alleys would branch from the standard path, and researchers would tally how many errors the animal committed by briefly turning into a blind (*Tolman and Honzik, 1930*). Later on, the design was further reduced to a single T-junction. After all, the elementary act of maze navigation is whether to turn left or right at a junction (*Tolman, 1938*), so why not study that process in isolation? And reducing the concept even further, one can ask the animal to refrain from walking altogether, and instead poke its nose into a hole on the left or the right side of a box (*Uchida and Mainen, 2003*). This led to the popular behavior boxes now found in rodent neuroscience laboratories everywhere. Each of these reductions of the 'maze' concept enabled a new type of experiment to study learning and decision-making, for example limiting the number of choice points allows one to better sample neural activity at each one. However, the essence of a 'confusing network of paths' has been lost along the way, and with it the behavioral richness of the animals navigating those decisions.

Owing in part to the dissemination of user-friendly tools for animal tracking, one sees a renaissance of experiments that embrace complex environments, including mazes with many choice points (*Alonso et al., 2020*; *Wood et al., 2018*; *Sato et al., 2018*; *Nagy et al., 2020*; *Rondi-Reig et al., 2006*; *Yoder et al., 2011*; *McNamara et al., 2014*), 3-dimensional environments (*Grobéty and Schenk, 1992*), and infinite mazes (*Shokaku et al., 2020*). The labyrinth in the present study is considerably more complex than Hampton Court or most of the mazes employed by Tolman and others (*Tolman and Honzik, 1930*; *Buel, 1934*; *Munn, 1950a*). In those mazes, the blind alleys are all short and unbranched; when an animal strays from the target path it receives feedback quickly and can correct. By contrast, our binary tree maze has 64 equally deep branches, only one of which contains the reward port. If the animal makes a mistake at any level of the tree, it can find out only after traveling all the way to the last node.

Another crucial aspect of our experimental design is the absence of any human interference. Most studies of animal navigation and learning involve some kind of trial structure. For example, the experimenter puts the rat in the start box, watches it make its way through the maze, coaxes it back on the path if necessary, and picks it up once it reaches the target box. Then another trial starts. In modern experiments with two-alternative-forced-choice (2AFC) behavior boxes the animal doesn't have to be picked up, but a trial starts with appearance of a cue, and then proceeds through some strict protocol through delivery of the reward. The argument in favor of imposing a trial structure is that it creates reproducible conditions, so that one can gather comparable data and average them suitably over many trials.

Our experiments had no imposed structure whatsoever; in fact, it may be inappropriate to call them experiments. The investigator opened the entry to the maze in the evening and did not return until the morning. A potential advantage of leaving the animals to themselves is that they are more likely to engage in mouse-like behavior, rather than constantly responding to the stress of human interference or the alienation from being a cog in a behavior machine. The result was a rich data set, with the typical animal delivering ~15,000 decisions in a single night, even if one only counts the nodes of the binary tree as decision points. Since the mice made all the choices, the scientific effort lay primarily in adapting methods of data analysis to the nature of mouse trajectories. Somewhat surprisingly, the absence of experimental structure was no obstacle to making precise and reproducible measurements of the animal's behavior.

## How fast do animals learn?

Among the wide range of phenomena of animal learning, one can distinguish easy and hard tasks by some measure of task complexity. In a simple picture of a behavioral task, the animal needs to recognize several different contexts and based on that express one of several different actions. One can draw up a contingency table between contexts and actions, and measure the complexity of the task by the mutual information in that table. This ignores any task difficulties associated with sensing the context at all or with producing the desired actions. However, in all the examples discussed here, the stimuli are discriminated easily and the actions come naturally, thus the learning difficulty lies only in forming the associations, not in sharpening the perceptual mechanisms or practicing complex motor output.

Many well-studied behaviors have a complexity of 1 bit or less, and often animals can learn these associations after a single experience. For example, in the Bruce effect (*Bruce, 1959*), the female maps two different contexts (smell of mate vs non-mate) onto two kinds of pregnancy outcomes

(carry to term vs abort). The mutual information in that contingency table is at most one bit, and may be considerably lower, for example if non-mate males are very rare or very frequent. Mice form the correct association after a single instance of mating, although proper memory formation requires several hours of exposure to the mate odor (*Rosser and Keverne, 1985*).

Similarly fear learning under the common electroshock paradigm establishes a mapping between two contexts (paired with shock vs innocuous) and two actions (freeze vs proceed), again with an upper bound of 1 bit of complexity. Rats and mice will form the association after a single experience lasting only seconds, and alter their behavior over several hours (*Fanselow and Bolles, 1979*; *Bourtchuladze et al., 1994*). This is an adaptive warning system to deal with life-threatening events, and rapid learning here has a clear survival value.

Animals are particularly adept at learning a new association between an odor and food. For example, bees will extend their proboscis in response to a new odor after just one pairing trial where the odor appeared together with sugar (*Bitterman et al., 1983*). Similarly, rodents will start digging for food in a scented bowl after just a few pairings with that odor (*Cleland et al., 2009*). Again, these are 1-bit tasks learned rapidly after one or a few experiences.

By comparison, the tasks that a mouse performs in the labyrinth are more complex. For example, the path from the maze entrance to the water port involves six junctions, each with three options. At a minimum six different contexts must be mapped correctly into one of three actions each, which involves $6 \cdot \log_2 3 = 9.5$ bits of complexity. The animals begin to execute perfect paths from the entrance to the water port well within the first hour (*Figure 2C*, *Figure 3B*). At a later stage during the night, the animal learns to walk direct paths to water from many different locations in the maze (*Figure 5*); by this time, it has consumed 10–20 rewards. In the limit, if the animal could turn correctly towards water from each of 63 junctions in the maze, it would have learned $63 \cdot \log_2 3 = 100$ bits. Conservatively, we estimate that the animals have mastered 10–20 bits of complexity based on 10–20 reward experiences within an hour of time spent in the maze. Note this considers only information about the water port and ignores whatever else the animals are learning about the maze during their incessant exploratory forays. These numbers align well with classic experiments on rats in diverse mazes and problem boxes *Munn, 1950a*. Although those tasks come in many varieties, a common theme is that ~10 successful trials are sufficient to learn ~10 decisions (*Woodrow, 1942*).

In a different corner of the speed-complexity space are the many 2-alternative-forced-choice (2AFC) tasks in popular use today. These tend to be 1-bit tasks, for example the monkey should flick its eyes to the left when visual motion is to the left (*Newsome and Paré, 1988*), or the mouse should turn a steering wheel to the right when a light appears on the left (*Burgess et al., 2017*). Yet, the animals take a long time to learn these simple tasks. For example, the mouse with the steering wheel requires about 10,000 experiences before performance saturates. It never gets particularly good, with a typical hit rate only 2/3 of the way from random to perfect. All this training takes 3–6 weeks; in the case of monkeys several months. The rate of learning, measured in task complexity per unit time, is surprisingly low: less than 1 bit/month compared to ~10 bits/h observed in the labyrinth. The difference is a factor of 6000. Similarly when measured in complexity learned per reward experience: The 2AFC mouse may need 5000 rewards to learn a contingency table with one bit complexity, whereas the mouse in the maze needs ~10 rewards to learn 10 bits. Given these enormous differences in learning rate, one wonders whether the ultra-slow mode of learning has any relevance for an animal's natural condition. In the month that the 2AFC mouse requires to finally report the location of a light, its relative in the wild has developed from a baby to having its own babies. Along the way, that wild mouse had to make many decisions, often involving high stakes, without the benefit of 10,000 trials of practice.

## Sudden insight

The dynamics of the learning process are often conceived as a continuously growing association between stimuli and actions, with each reinforcing experience making an infinitesimal contribution. The reality can be quite different. When a child first learns to balance on a bicycle, performance goes from abysmal to astounding within a few seconds. The timing of such a discontinuous step in performance seems impossible to predict but easy to recognize after the fact.

From the early days of animal learning experiments, there have been warnings against the tendency to average learning curves across subjects (*Krechevsky, 1932*; *Estes, 1956*). The average of many discontinuous curves will certainly look continuous and incremental, but that reassuring shape

may miss the essence of the learning process. A recent reanalysis of many Pavlovian conditioning experiments suggested that discontinuous steps in performance are the rule rather than the exception (*Gallistel et al., 2004*). Here, we found that the same applies to navigation in a complex labyrinth. While the average learning curve presents like a continuous function (*Figure 3B*), the individual records of water rewards show that each animal improves rather quickly but at different times (*Figure 3A*).

Owing to the unstructured nature of the experiment, the mouse may adopt different policies for getting to the water port. In at least half the animals, we observed a discontinuous change in that policy, namely when the animal started using efficient direct paths within the maze (*Figure 5*, *Figure 5—source data 1*). This second switch happened considerably after the animal started collecting rewards, and did not greatly affect the reward rate. Furthermore, the animals never reverted to the less efficient policy, just as a child rarely unlearns to balance a bicycle.

Presumably, this switch in performance reflects some discontinuous change in the animal's internal model of the maze, what Tolman called the 'cognitive map' (*Tolman, 1948*; *Behrens et al., 2018*). In the unrewarded animals, we could not detect any discontinuous change in the use of long paths. However, as Tolman argued, those animals may well acquire a sophisticated cognitive map that reveals itself only when presented with a concrete task, like finding water. Future experiments will need to address this. The discontinuous changes in performance pose a challenge to conventional models of reinforcement learning, in which reward events are the primary driver of learning and each event contributes an infinitesimal update to the action policy. It will also be important to model the acquisition of distinct kinds of knowledge that contribute to the same behavior, like the location of the target and efficient routes to approach it.

## Exploratory behavior

By all accounts, the animals spent a large fraction of the night exploring the maze (*Figure 1—figure supplement 2*). The water-deprived animals continued their forays into the depths of the maze long after they had found the water port and learned to exploit it regularly. After consuming a water reward, they wandered off into the maze 90% of the time (*Figure 7B*) instead of lazily waiting in front of the port during the timeout period. The sated animals experienced no overt reward from the maze, yet they likewise spent nearly half their time exploring that environment. As has been noted many times, animals – like humans – derive some form of intrinsic reward from exploration (*Berlyne, 1960*). Some have suggested that there exists a homeostatic drive akin to hunger and thirst that elicits the information-seeking activity, and that the drive is in turn sated by the act of exploration (*Hughes, 1997*). If this were the case then the drive to explore should be weakest just after an episode of exploration, much as the drive for food-seeking is weaker after a big meal.

Our observations are in conflict with this notion. The animal is most likely to enter the maze within the first minute of its return to the cage (*Figure 1—figure supplement 3*), a strong trend that runs opposite to the prediction from satiation of curiosity. Several possible explanations come to mind: (1) On these very brief visits to the cage the animal may just want to certify that the exit route to the safe environment still exists, before continuing with exploration of the maze. (2) The temporal contrast between the boredom of the cage and the mystery of the maze is highest right at the moment of exit from the maze, and that may exert pressure to re-enter the maze. Understanding this in more detail will require dedicated experiments. For example, one could deliberately deprive the animals of access to the maze for some hours, and test whether that results in an increased drive to explore, as observed for other homeostatic drives around eating, drinking, and sleeping.

When left to their own devices, mice choose to spend much of their time engaged in exploration. One wonders how that affects their actions when they are strapped into a rigid behavior machine, like a 2AFC choice box. Presumably the drive to explore persists, perhaps more so because the forced environment is so unpleasant. And within the confines of the two alternatives, the only act of exploration the mouse has left is to give the wrong answer. This would manifest as an unexpectedly high error rate on unambiguous stimuli, sometimes called the 'lapse rate' (*Carandini and Churchland, 2013*; *Pisupati et al., 2021*). The fact that the lapse rate decreases only gradually over weeks to months of training (*Burgess et al., 2017*) suggests that it is difficult to crush the animal's drive to explore.

The animals in our experiments had never been presented with a maze environment, yet they quickly settled into a steady mode of exploration. Once a mouse progressed beyond the first

intersection it typically entered deep into the maze to one or more end nodes (*Figure 6*). Within 50 s of the first entry, the animals adopted a steady speed of locomotion that they would retain throughout the night (*Figure 2—figure supplement 1*). Within 250 s of first contact with the maze, the average animal already spent 50% of its time there (*Figure 1—figure supplement 2*). Contrast this with a recent study of 'free exploration' in an exposed arena: Those animals required several hours before they even completed one walk around the perimeter (*Fonio et al., 2009*). Here the drive to explore is clearly pitted against fear of the open space, which may not be conducive to observing exploration per se.

The persistence of exploration throughout the entire duration of the experiment suggests that the animals are continuously surveying the environment, perhaps expecting new features to arise. These surveys are quite efficient: The animals cover all parts of the maze much faster than expected from a random walk (*Figure 8*). Effectively they avoid re-entering territory they surveyed just recently. It is often assumed that this requires some global memory of places visited in the environment (*Nagy et al., 2020*; *Olton, 1979*). Such memory would have to persist for a long time: Surveying half of the available end nodes typically required 450 turning decisions. However, we found that a global long-term memory is not needed to explain the efficient search. The animals seem to be governed by a set of local turning biases that require memory only of the most recent decision and no knowledge of location (*Figure 9*). These local biases alone can explain most of the character of exploration without any global understanding or long-term memory. Incidentally, they also explain other seemingly global aspects of the behavior, for example the systematic preference that the mice have for the outer rather than the inner regions of the maze (*Figure 10*). Of course, this argument does not exclude the presence of a long-term memory, which may reveal itself in some other feature of the behavior.

Perhaps, the most remarkable aspect of these biases is how similar they are across all 19 mice studied here, regardless of whether the animal experienced water rewards or not (*Figure 9B*, *Figure 9—source data 1*), and independent of the sex of the mouse. The four decision probabilities were identical across individuals to within a standard deviation of less than 0.03. We cannot think of a trivial reason why this should be so. For example the two biases for forward motion (*Figure 9B* left) are poised halfway between the value for a random walk ($p = 2/3$) and certainty ($p = 1$). At either of those extremes, simple saturation might lead to a reproducible value, but not in the middle of the range. Why do different animals follow the exact same decision rules at an intersection between tunnels? Given that tunnel systems are part of the mouse's natural ecology, it is possible that those rules are innate and determined genetically. Indeed the rules by which mice build tunnels have a strong genetic component (*Weber et al., 2013*), so the rules for using tunnels may be written in the genes as well. The high precision with which one can measure those behaviors even in a single night of activity opens the way to efficient comparisons across genotypes, and also across animals with different developmental experience.

Finally, after mice discover the water port and learn to access it from many different points in the maze (*Figure 5*), they are presumably eager to discover other things. In ongoing work, we installed three water ports (visible in the videos accompanying this article) and implemented a rule that activates the three ports in a cyclic sequence. Mice discovered all three ports rapidly and learned to visit them in the correct order. Future experiments will have to raise the bar on what the mice are expected to learn in a night.

## Mechanisms of navigation

How do the animals navigate when they perform direct paths to the water port or to the exit? The present study cannot resolve that, but one can gain some clues based on observations so far. Early workers already concluded that rodents in a maze will use whatever sensory cues and tricks are available to accomplish their tasks (*Munn, 1950b*). Our maze was designed to restrict those options somewhat.

To limit the opportunity for visual navigation, the floor and walls of the maze are visually opaque. The ceiling is transparent, but the room is kept dark except for infrared illuminators. Even if the animal finds enough light, the goals (water port or exit) are invisible within the maze except from the immediately adjacent corridor. There are no visible beacons that would identify the goal.

With regard to the sense of touch and kinesthetics, the maze was constructed for maximal symmetry. At each level of the binary tree, all the junctions have locally identical geometry, with

intersecting corridors of the same length. In practice, the animals may well detect some inadvertent cues, like an unusual drop of glue, that could identify one node from another. The maze rotation experiment suggests that such cues are not essential for the animal's sense of location in the maze, at least in the expert phase.

The role of odors deserves particular attention because the mouse may use them both passively and actively. Does the animal first find the water port by following the smell of water? Probably not. For one, the port only emits a single drop of water when triggered by a nose poke. Second, we observed many instances where the animal is in the final corridor adjacent to the water port yet fails to discover it. The initial discovery seems to occur via touch. The reader can verify this in the videos accompanying this article. Regarding active use of odor markings in the maze, the maze rotation experiment suggests that such cues are not required for navigation, at least once the animals have adopted the shortest path to the water port (*Figure 4*).

Another algorithm that is often invoked for animals moving in an open arena is vector-based navigation (*Wehner et al., 1996*). Once the animal discovers a target, it keeps track of that target's heading and distance using a path integrator. When it needs to return to the target it follows the heading vector and updates heading and distance until it arrives. Such a strategy has limited appeal inside a labyrinth because the vectors are constantly blocked by walls. Consider, for example, the 'home runs' back to the exit at the end of a bout. Here the target, namely the exit, is known from the start of the bout, because the animal enters through the same hole. At the end of the bout, when the mouse decides to exit from the maze, can it follow the heading vector to the exit? *Figure 6A* shows the 13 locations from which mice returned in a direct path to the exit on their very first foray. None of these locations is compatible with heading-based navigation: In each case an animal following the heading to the exit would get stuck in a different end node first and would have to reverse from there, quite unlike what really happened.

Finally, a partial clue comes from errors the animals make. We found that the rotation image of the water port, an end node diametrically across the entire maze, is one of the most popular destinations for rewarded animals (*Figure 10A*). These errors would be highly unexpected if the animals navigated from the entrance to the water by odor markings, or if they used an absolute representation of heading and distance. On the other hand, if the animal navigates via a remembered sequence of turns, then it will end up at that image node if it makes a single mistake at just the first T-junction.

Future directed experiments will serve to narrow down how mice learn to navigate this environment, and how their policy might change over time. Since the animals get to perfection within an hour or so, one can test a new hypothesis quite efficiently. Understanding what mechanisms they use will then inform thinking about the algorithm for learning, and about the neuronal mechanisms that implement it.

## Materials and methods

### Experimental design

The goal of the study was to observe mice as they explored a complex environment for the first time, with little or no human interference and no specific instructions. In preliminary experiments, we tested several labyrinth designs and water reward schedules. Eventually, we settled on the protocol described here, and tested 20 mice in rapid succession. Each mouse was observed only over a 7-h period during the first night it encountered the labyrinth.

### Maze construction

The maze measured ~24 x 24 x 2 inches; for manufacture we used materials specified in inches, so dimensions are quoted in those non-SI units where appropriate. The ceiling was made of 0.5 inch clear acrylic. Slots of 1/8 inch width were cut into this plate on a 1.5 inch grid. Pegged walls made of 1/8 inch infrared-transmitting acrylic (opaque in the visible spectrum, ePlastics) were inserted into these slots and secured with a small amount of hot glue. The floor was a sheet of infrared-transmitting acrylic, supported by a thicker sheet of clear acrylic. The resulting corridors (1-1/8 inches wide) formed a 6-level binary tree with T-junctions and progressive shortening of each branch, ranging from ~12 inch to 1.5 inch (*Figure 1* and *Figure 2*). A single end node contained a 1.5 cm circular

opening with a water delivery port (described below). The maze included provision for two additional water ports not used in the present report. Once per week the maze was submerged in cage cleaning solution. Between different animals the floor and walls were cleaned with ethanol.

## Reward delivery system

The water reward port was controlled by a Matlab script on the main computer through an interface (Sanworks Bpod State Machine r1). Rewards were triggered when the animal's nose broke the IR beam in the water port (Sanworks Port interface + valve). The interface briefly opened the water valve to deliver ~30 μL of water and flashed an infrared LED mounted outside the maze for 1 s. This served to mark reward events on the video recording. Following each reward, the system entered a time-out period for 90 s, during which the port did not provide further reward. In experiments with sated mice the water port was turned off.

## Cage and connecting passage

The entrance to the maze was connected to an otherwise normal mouse cage by red plastic tubing (3 cm dia, 1 m long). The cage contained food, bedding, nesting material, and in the case of unrewarded experiments also a normal water bottle.

## Animals and treatments

All mice were C57BL/6J animals (Jackson Labs) between the ages of 45 and 98 days (mean 62 days). Both sexes were used: four males and six females in the rewarded experiments, five males and four females in the unrewarded experiments. For water deprivation, the animal was transferred from its home cage (generally group-housed) to the maze cage ~22 h before the start of the experiment. Non-deprived animals were transferred minutes before the start. All procedures were performed in accordance with institutional guidelines and approved by the Caltech IACUC.

## Video recording

All data reported here were collected over the course of 7 h during the dark portion of the animal's light cycle. Video recording was initiated a few seconds prior to connecting the tunnel to the maze. Videos were recorded by an OpenCV python script controlling a single webcam (Logitech C920) located ~1 m below the floor of the maze. The maze and access tube were illuminated by multiple infrared LED arrays (center wavelength 850 nm). Three of these lights illuminated the maze from below at a 45 degree angle, producing contrast to resolve the animal's foot pads. The remaining lights pointed at the ceiling of the room to produce backlight for a sharp outline of the animal.

## Animal tracking

A version of DeepLabCut (*Nath et al., 2019*) modified to support gray-scale processing was used to track the animal's trajectory, using key points at the nose, feet, tail base and mid-body. All subsequent analyses were based on the trajectory of the animal's nose, consisting of positions $x(t)$ and $y(t)$ in every video frame.

## Rates of transition between cage and maze

This section relates to *Figure 1—figure supplement 3*. We entertained the hypothesis that the animals become 'thirsty for exploration' as they spend more time in the cage. In that case one would predict that the probability of entering the maze in the next second will increase with time spent in the cage. One can compute this probability from the distribution of residency times in the cage, as follows:

Say $t = 0$ when the animal enters the cage. The probability density that the animal will next leave the cage at time $t$ is

$$p(t) = e^{-\int_0^t r(t')dt'} r(t) \qquad (2)$$

where $r(t)$ is the instantaneous rate for entering the maze. So

$$\int_0^t p(t')dt' = 1 - e^{-\int_0^t r(t')dt'}\tag{3}$$

$$\int_0^t r(t')dt' = -\ln\left(1 - \int_0^t p(t')dt'\right)\tag{4}$$

This relates the cumulative of the instantaneous rate function to the cumulative of the observed transition times. In this way we computed the rates

$$r_{\mathrm{m}}(t) = \text{rate of entry into the maze as a function of time spent in the cage}\tag{5}$$

$$r_{\mathrm{c}}(t) = \text{rate of entry into the cage as a function of time spent in the maze}\tag{6}$$

The rate of entering the maze is highest at short times in the cage (**Figure 1—figure supplement 3A**). It peaks after ~15 s in the cage and then declines gradually by a factor of 4 over the first minute. So the mouse is most likely to enter the maze just after it returns from there. This runs opposite to the expectation from a homeostatic drive for exploration, which should be sated right after the animal returns. We found no evidence for an increase in the rate at late times. These effects were very similar in rewarded and unrewarded groups and in fact the tendency to return early was seen in every animal.

By contrast, the rate of exiting the maze is almost perfectly constant over time (**Figure 1—figure supplement 3B**). In other words, the exit from the maze appears like a constant rate Poisson process. There is a slight elevation of the rate at short times among rewarded animals (**Figure 1—figure supplement 3B** top). This may come from the occasional brief water runs they perform. Another strange deviation is an unusual number of very short bouts (duration 2–12 s) among unrewarded animals (**Figure 1—figure supplement 3B** bottom). These are brief excursions in which the animal runs to the central junction, turns around, and runs to the exit. Several animals exhibited these, often several bouts in a row, and at all times of the night.

## Reduced trajectories

From the raw nose trajectory we computed two reduced versions. First, we divided the maze into discrete 'cells', namely the squares the width of a corridor that make up the grid of the maze. At any given time, the nose is in one of these cells and that time series defines the cell trajectory.

At a coarser level still one can ask when the animal passes through the nodes of the binary tree, which are the decision points in the maze. The special cells that correspond to the nodes of the tree are those at the center of a T-junction and those at the leaves of the tree. We marked all the times when the trajectory $(x(t), y(t))$ entered a new node cell. If the animal leaves a node cell and returns to it before entering a different node cell, that is not considered a new node. This procedure defines a discrete node sequence $s_i$ and corresponding arrival times at those nodes $t_i$. We call the transition between two nodes a 'step'. Much of the analysis in this paper is derived from the animal's node sequence. The median mouse performed 16,192 steps in the 7 h period of observation (mean = 15,257; SD = 3340).

In **Figure 5** and **Figure 6**, we count the occurrence of direct paths leading to the water port (a 'water run') or to the exit (a 'home run'). A direct path is a node sequence without any reversals. **Figure 3—figure supplement 1** illustrates some examples.

If the animal makes one wrong step from the direct path, that step needs to be backtracked, adding a total of two steps to the length of the path. If further errors occur during backtracking, they need to be corrected as well. The binary maze contains no loops, so the number of errors is directly related to the length of the path:

$$\text{Errors} = (\text{Length of path} - \text{Length of direct path})/2\tag{7}$$

## Maze rotation

The maze rotation experiment (*Figure 4*) was performed on four mice, all water-deprived. Two of the animals ('D7' and 'D9') had experienced the maze before, and are part of the 'rewarded' group in other sections of the report. Two additional animals ('F2' and 'A1') had had no prior contact with the maze.

The maze rotation occurred after at least 6 h of exposure, by which time the animals had all perfected the direct path to the water port.

For animals 'D7' and 'D9' we rotated only the floor of the maze, leaving the walls and ceiling in the original configuration. For 'F2' and 'A1' we rotated the entire maze, moving one wall segment at the central junction and the water port to attain the same shape. Navigation remained intact for all animals. Note that 'A1' performed a perfect path to the water port and back immediately before and after a full maze rotation (*Figure 4B*).

The visits to the four locations in the maze (*Figure 4C*, *Figure 4—figure supplement 1*) were limited to direct paths of length at least two steps. This avoids counting rapid flickers between two adjacent nodes. In other words, the animal has to move at least two steps away from the target node before another visit qualifies.

## Statistics of sudden insight

In *Figure 5* one can distinguish two events: First, the animal finds the water port and begins to collect rewards at a steady rate: this is when the green curve rises up. At a later time, the long direct paths to the water port become much more frequent than to the comparable control nodes: this is when the red and blue curves diverge. For almost all animals these two events are well separated in time (*Figure 5—figure supplement 1*). In many cases, the rate of long paths seems to change discontinuously: a sudden change in slope of the curve.

Here, we analyze the degree of 'sudden change', namely how rapidly the rate changes in a time series of events. We modeled the rate as a sigmoid function of time during the experiment:

$$r(t) = r_\mathrm{i} + \frac{r_\mathrm{f} - r_\mathrm{i}}{2} \operatorname{erf}\left(\frac{t - t_\mathrm{s}}{w}\right) \tag{8}$$

where

$$\operatorname{erf}(x) = \frac{2}{\sqrt{\pi}} \int_0^x e^{-x^2} \mathrm{d}x$$

The rate begins at a low initial level $r_\mathrm{i}$, reflecting chance occurrence of the event, and saturates at a high final level $r_\mathrm{f}$, limited for example by the animal's walking speed. The other two parameters are the time $t_\mathrm{s}$ of half-maximal rate change, and the width $w$ over which that rate change takes place. A sudden change in the event rate would correspond to $w = 0$.

The data are a set of $n$ event times $t_i$ in the observation interval $[0, T]$. We model the event train as an inhomogeneous Poisson point process with instantaneous rate $r(t)$. The likelihood of the data given the rate function $r(t)$ is

$$L[r(t)] = e^{-\int_0^T r(t)dt} \prod_i r(t_i) \tag{9}$$

and the log likelihood is

$$\ln L = \sum_i \ln r(t_i) - \int_0^T r(t)dt \tag{10}$$

For each of the 10 rewarded mice, we maximized $\ln L$ over the 4 parameters of the rate model, both for the reward events and the long paths to water. The resulting fits are plotted in *Figure 5—figure supplement 1*.

Focusing on the learning of long paths to water, for 6 of the 10 animals the optimal width parameter $w$ was less than 300 s: B1, B2, C1, C3, C6, C7. These are the same animals one would credit

with a sudden kink in the cumulative event count based on visual inspection (*Figure 5—figure supplement 1*).

To measure the uncertainty in the timing of this step, we refit the data for this subgroup of mice with a model involving a sudden step in the rate,

$$r(t) = \begin{cases} r_i, & t < t_s \\ r_f, & t > t_s \end{cases}$$

(11)

and computed the likelihood of the data as a function of the step time $t_s$. We report the mean and standard deviation of the step time over its likelihood in *Figure 5—source data 1*. Animal C6 was dropped from this 'sudden step' group, because the uncertainty in the step time was too large (~900 s).

## Efficiency of exploration

The goal of this analysis is to measure how effectively the animal surveys all the end nodes of the maze. The specific question is: In a string of $n$ end nodes that the animal samples, how many of these are distinct? On average how does the number of distinct nodes $d$ increase with $n$? This was calculated as follows:

We restricted the animal's node trajectory ($s_i$) to clips of exploration mode, excluding the direct paths to the water port or the exit. All subsequent steps were applied to these clips, then averaged over clips. Within each clip we marked the sequence of end nodes ($e_i$). We slid a window of size $n$ across this sequence and counted the number of distinct nodes $d$ in each window. Then we averaged $d$ over all windows in all clips. Then we repeated that for a wide range of $n$. The resulting $d(n)$ is plotted in the figures reporting new nodes vs nodes visited (*Figure 8A,B* and *Figure 9C*).

For a summary analysis, we fitted the curves of $d(n)$ with a two-parameter function:

$$d(n) \approx 64 \left( 1 - \frac{1}{1 + \frac{z + bz^3}{1 + b}} \right)$$

(12)

where

$$z = n/a$$

(13)

The parameter $a$ is the number of visits $n$ required to survey half of the end nodes, whereas $b$ reflects a relative acceleration in discovering the last few end nodes. This function was found by trial and error and produces absurdly good fits to the data (*Figure 8—figure supplement 1*). The values quoted in the text for efficiency of exploration are $E = 32/a$ (*Equation 1*).

The value of $b$ was generally small (~0.1) with no difference between rewarded and unrewarded animals. It declined slightly over the night (*Figure 8—figure supplement 1B*), along with the decline in $a$ (*Figure 8C*).

## Biased random walk

For the analysis of *Figure 9*, we considered only the parts of the trajectory during 'exploration' mode. Then we parsed every step between two nodes in terms of the type of action it represents. Note that every link between nodes in the maze is either a 'left branch' or a 'right branch', depending on its relationship to the parent T-junction. Therefore, there are four kinds of action:

- $a = 0$: 'in left', take a left branch into the maze
- $a = 1$: 'in right', take a right branch into the maze
- $a = 2$: 'out left', take a left branch out of the maze
- $a = 3$: 'out right', take a right branch out of the maze

At any given node, some actions are not available, for example from an end node one can only take one of the 'out' actions.

To compute the turning biases, we considered every T-junction along the trajectory and correlated the action $a_0$ that led into that node with the subsequent action $a_1$. By tallying the action pairs $(a_0, a_1)$, we computed the conditional probabilities $p(a_1|a_0)$. Then the four biases are defined as

$$P_{\mathrm{SF}} = \frac{p(0|0)+p(0|1)+p(1|0)+p(1|1)}{p(0|0)+p(0|1)+p(1|0)+p(1|1)+p(2|0)+p(3|1)} \tag{14}$$

$$P_{\mathrm{SA}} = \frac{p(0|1)+p(1|0)}{p(0|0)+p(0|1)+p(1|0)+p(1|1)} \tag{15}$$

$$P_{\mathrm{BF}} = \frac{p(0|3)+p(1|2)+p(2|2)+p(2|3)+p(3|2)+p(3|3)}{p(0|3)+p(1|2)+p(2|2)+p(2|3)+p(3|2)+p(3|3)+p(0|2)+p(1|3)} \tag{16}$$

$$P_{\mathrm{BS}} = \frac{p(2|2)+p(2|3)+p(3|2)+p(3|3)}{p(0|3)+p(1|2)+p(2|2)+p(2|3)+p(3|2)+p(3|3)} \tag{17}$$

For the simulations of random agents (*Figure 8*, *Figure 9*), we used trajectories long enough so the uncertainty in the resulting curves was smaller than the line width.

## Models of decisions during exploration

The general approach is to develop a model that assigns probabilities to the animal's next action, namely which node it will move to next, based on its recent history of actions. All the analyses were restricted to the animal's 'exploration' mode and to the 63 nodes in the maze that are T-junctions. During the 'drink' and 'leave' modes the animal's next action is predictable. Similarly, when it finds itself at one of the 64 end nodes it only has one action available.

For every mouse trajectory, we split the data into five segments, trained the model on 80% of the data, and tested it on 20%, averaging the resulting cross-entropy over the five possible splits. Each segment was in turn composed of parts of the trajectory sampled evenly throughout the 7 h experiment, so as to average over the small changes in the course of the night. The model was evaluated by the cross-entropy between the predictions and the animal's true actions. If one had an optimal model of behavior, the result would reveal the animal's true source entropy.

### Fixed depth Markov chain

To fit a model with fixed history depth $k$ to a measured node sequence $(s_t)$, we evaluated all the substrings in that sequence of length $(k+1)$. At any given time $t$, the $k$-string $\mathbf{h}_t = (s_{t-k+1}, \ldots, s_t)$ identifies the history of the animal's $k$ most recent locations. The current state $s_t$ is one of 63 T-junctions. Each state is preceded by one of 3 possible states. So the number of history strings is $63 \cdot 3^{k-1}$. The 2-string $(s_t, s_{t+1})$ identifies the next action $a_{t+1}$, which can be 'in left', 'in right', or 'out', corresponding to the 3 branches of the T junction. Tallying the history strings with the resulting actions leads to a contingency table of size $63 \cdot 3^{k-1} \times 3$, containing

$$n(\mathbf{h}, a) = \text{number of times history } \mathbf{h} \text{ leads to action a} \tag{18}$$

Based on these sample counts, we estimated the probability of each action $a$ conditional on the history $\mathbf{h}$ as

$$p(a|\mathbf{h}) = \frac{n(\mathbf{h}, a) + 1}{\sum_{a'} n(\mathbf{h}, a') + 3} \tag{19}$$

This amounts to additive smoothing with a pseudocount of 1, also known as 'Laplace smoothing'. These conditional probabilities were then used in the testing phase to predict the action at time $t$ based on the preceding history $\mathbf{h}_t$. The match to the actually observed actions $a_t$ was measured by the cross-entropy

$$H = \langle -\log_2 p(a_t|\mathbf{h}_t) \rangle_t \tag{20}$$

### Variable depth Markov chain

As one pushes to longer histories, that is larger $k$, the analysis quickly becomes data-limited, because the number of possible histories grows exponentially with $k$. Soon one finds that the counts for each

history-action combination drop to where one can no longer estimate probabilities correctly. In an attempt to offset this problem, we pruned the history tree such that each surviving branch had more than some minimal number of counts in the training data. As expected, this model is less prone to over-fitting and degrades more gently as one extends to longer histories (*Figure 11—figure supplement 1A*). The lowest cross-entropy was obtained with an average history length of ~4.0 but including some paths of up to length 6. Of all the algorithms we tested, this produced the lowest cross-entropies, although the gains relative to the fixed-depth model were modest (*Figure 11—figure supplement 1C*).

### Pooling across symmetric nodes in the maze

Another attempt to increase the counts for each history involved pooling counts over multiple T-junctions in the maze that are closely related by symmetry. For example, all the T-junctions at the same level of the binary tree look locally similar, in that they all have corridors of identical length leading from the junction. If one supposes that the animal acts the same way at each of those junctions, one would be justified in pooling across these nodes, leading to a better estimate of the action probabilities, and perhaps less over-fitting. This particular procedure was unsuccessful, in that it produced higher cross-entropy than without pooling.

However, one may want to distinguish two types of junctions within a given level: L-nodes are reached by a left branch from their parent junction one level lower in the tree, R-nodes by a right branch. For example, in *Figure 3—figure supplement 1*, node 1 is L-type and node 2 is R-type. When we pooled histories over all the L-nodes at a given level and separately over all the R-nodes the cross-entropy indeed dropped, by about 5% on average. This pooling greatly reduced the amount of over-fitting (*Figure 11—figure supplement 1B*), which allowed the use of longer histories, which in turn improved the predictions on test data. The benefit of distinguishing L- and R-nodes probably relates to the animal's tendency to alternate left and right turns.

All the Markov model results we report are obtained using pooling over L-nodes and R-nodes at each maze level.

## Acknowledgements

Funding: This work was supported by the Simons Collaboration on the Global Brain (grant 543015 to MM and 543025 to PP), by NSF award 1564330 to PP, and by a gift from Google to PP.

## Additional information

### Competing interests

Markus Meister: Reviewing editor, *eLife*. The other authors declare that no competing interests exist.

### Funding

| Funder | Grant reference number | Author |
| --- | --- | --- |
| Simons Foundation | 543015 | Markus Meister |
| Simons Foundation | 543025 | Pietro Perona |
| National Science Foundation | 1564330 | Pietro Perona |
| Google | | Pietro Perona |

The funders had no role in study design, data collection and interpretation, or the decision to submit the work for publication.

### Author contributions

Matthew Rosenberg, Tony Zhang, Conceptualization, Data curation, Software, Formal analysis, Validation, Investigation, Visualization, Methodology, Writing - review and editing; Pietro Perona, Conceptualization, Software, Formal analysis, Supervision, Funding acquisition, Validation, Visualization,

Methodology, Project administration, Writing - review and editing; Markus Meister, Conceptualization, Software, Formal analysis, Supervision, Funding acquisition, Validation, Visualization, Methodology, Writing - original draft, Project administration, Writing - review and editing

### Author ORCIDs
Tony Zhang https://orcid.org/0000-0002-5198-499X
Markus Meister https://orcid.org/0000-0003-2136-6506

### Ethics
Animal experimentation: This study was performed in strict accordance with the recommendations in the Guide for the Care and Use of Laboratory Animals of the National Institutes of Health. All of the animals were handled according to animal protocol 1656 approved by the institutional animal care and use committee (IACUC) at Caltech.

### Decision letter and Author response
Decision letter https://doi.org/10.7554/eLife.66175.sa1
Author response https://doi.org/10.7554/eLife.66175.sa2

## Additional files

### Supplementary files
• Transparent reporting form

### Data availability
The behavioral data and code that produced the figures are available in a public Github repository cited in the article https://github.com/markusmeister/Rosenberg-2021-Repository (copy archived at https://archive.softwareheritage.org/swh:1:rev:224141473e53d6e8963a77fbe625f570b0903ef1). We also prepared a permanent institutional repository at https://data.caltech.edu/badge/latestdoi/329740227.

The following dataset was generated:

| Author(s) | Year | Dataset title | Dataset URL | Database and Identifier |
|---|---|---|---|---|
| Rosenberg M, Zhang T, Perona P, Meister M | 2021 | Rosenberg-2021-Repository | https://github.com/markusmeister/Rosenberg-2021-Repository | Github, markusmeister/Rosenberg-2021-Repository |

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
