## [Decision Letter]

Thank you for submitting your article "Mice in a labyrinth: Rapid learning, sudden insight, and efficient exploration" for consideration by *eLife*. Your article has been reviewed by 3 peer reviewers, and the evaluation has been overseen by Mackenzie Mathis as the Reviewing Editor and Catherine Dulac as the Senior Editor. The following individual involved in review of your submission has agreed to reveal their identity: Gordon J Berman (Reviewer #1).

Essential revisions:

– The role of odors in task performance. We would like to ask the authors to clarify their thinking and data with respect to the role of odors in the behavior. The manuscript as currently written gives the impression that odors play a negligible role, and that mouse behavior is comprised of a series of decisions internal to the brain (e.g., the use of the phrasing "sudden insight"; the abstract's statement that a mouse "executes correct 10-bit choices after only 10 reward experiences – a learning rate 1000-fold higher than in 2AFC experiments"; etc.) But it was not clear that odors play a negligible role.

First, the mention of odors at the end of the discussion was a little confusing: although the text as written seems to suggest that the effect of odors was negligible, the data described seems to indicate the very opposite conclusion. The authors mention an experiment in which they rotate the maze, to ask whether animals are following a previously laid down odor trail, and they write that "Following that rotation the animal did in fact make a few visits to the rotated port location." Don't those first few visits to the rotated location suggest that indeed, the animal *was* following an odor trail? Why was behavior in those first few visits apparently discarded as inconsequential? It would be helpful if the authors clarified their thinking here. The authors proceeded to write that the mice "then quickly exploited the real water port again, without any apparent re-learning." But couldn't that same odor trail be used as a scaffold for rapid behavior change in runs subsequent to the first few ones?

Second, the data for the odor experiments mentioned in those final discussion paragraphs should be shown and included as part of the manuscript (both the maze rotation data and the open water bowl data). Odors are a very significant potential confound to how these data are being interpreted. Allowing readers to see the data and evaluate it for themselves is important.

Third, we would like to invite the authors to consider and comment on the following and other odor-guided-behavior possibilities. This one seems, at first sight, to be consistent with much of their data. Consider a mouse that, as it runs it brushes against the walls, leaving behind a faint odor trail (it doesn't need to be urine or feces), and that tends to not reverse unless it reaches a dead end, but has a little bit of a tendency (doesn't need to be absolute) to avoid, based on odor, its own recent previous path. Wouldn't that mouse do a level 6 home run in its very first bout, with little overlap between entry and exit paths? Wouldn't it produce a trail very much like the trail shown in Figure 2A? Yet this mouse wouldn't have learnt *anything* internally, in its brain, about what constitutes a home path – the "learning" for this mouse would all have been in the physical odor trail it left behind. That is only one possibility for how odor could affect behavior. There could be myriad others. After an hour, the maze is going to have feces and urine in multiple spots. As written, the claims about sudden insights, without considering whether such insight moments correlate with sudden odorant depositions (which would seem likely to act as highly prominent landmarks for the animal), could give readers the impression that the data demonstrate that it's all about internal mental processes. We are not yet convinced that odors can be so thoroughly ruled out.

Fortunately, this can all be addressed in the writing. Whether or not mice use odors in the maze, the data is fascinating and the paper extremely interesting. We would invite the authors to consider addressing the issue from the beginning, in the introduction, so that readers can keep the issue in mind as they absorb the paper. (We phrase this as an invitation in the sense that this is a suggestion for the authors, not a mandatory requirement.) For example, a simple sentence in the introduction saying something like "preliminary control experiments suggested to us that odors did not play a major role in how mice learned and solved these navigation problems, but we did not fully control for odors and their potential role remains to be fully elucidated," would not detract from the work in any way, and would go a very long way towards making sure that, even as they are making their way through the paper, readers don't get the impression that the issue is settled and odors can be forgotten.

– We would ask the authors to consider the implications of having an imposed 90-second delay between rewards. The authors write that animals tend to explore the maze, as opposed to simply focusing on the water reward location (line 118; Figure 6 and accompanying main text). That the imposed 90-second delay could play a role in this is, surprisingly, not mentioned or considered. The 90-second delay also has an important implication for the "learning insight" analysis, as described in the next weakness below.

– Another issue is in regard to a central claim in the manuscript, which is that animals have sudden insight moments in which they learn the path to the water reward. A plot based on data from one animal is presented in the main text (Figure 4B), appearing to the eye to support the claim (green and red lines have a kink in them). But for unexplained reasons data from other animals is relegated to the supplementary information (Figure 4 supplement 1), and many of these other animals do not, to the eye, support the claim (lines are smoothly curved, not kinked). The quantitative analysis that is used to adjudicate between sudden insights (a sudden step in the reward rate) versus gradual learning (slow continuous rise in reward rate) is flawed and, in our view, cannot be relied on: the approach compares a step model versus a smooth 2nd-order (quadratic) curve for how reward rate changes over time. However, because there is an imposed 90-second delay between successive rewards, the reward rate is capped at a maximum of 40 per hour, and cannot grow beyond that. A quadratic curve with a positive second-order term would rise indefinitely and will therefore be a poor fit to the data, not because the animals don't learn in a smooth way, but because of the 90-second delay, thus artifactually biasing the results of the analysis towards the step model.

– We suggest that the authors show, in simulated data, how the 90 s delay would affect the quadratic ramp model to see how much of a straw man it is. While the authors are clear that they see discontinuities in only 5 out of the 10 water-deprived animals, perhaps it would be ideal for them to show a couple of the less-discrete learners in the main text for additional clarity.

– "One shot learning." We don't think the authors have convincingly shown true one-shot homing behavior. It is an interesting and important claim in the manuscript ("one-shot learning of the home path" is the title of a section), but it is not yet completely clear that the data support that interpretation. The authors' point is that "home runs" start out longer than what random gradual exploration would predict, and that point is indeed supported by the data. But for the authors' claims about one-shot learning of home runs "from locations not previously encountered" to be supported, it is critical that the first and second bouts have little overlap between entry and exit paths. This should be shown in a main text figure, but it isn't, and moreover, Figure 5-supplement does not show initial individual runs, but rather averages over all bouts. Thus, the critical information about the first trajectories is nowhere in the current manuscript. We ask the authors to clearly show these initial-runs data for the individual mice. We do think Figure 2A suggests that their claim about path overlap in initial bouts may actually be true, but this should be demonstrated. Otherwise, we suggest the claim of one-shot learning be toned down.

---

## [Author Response]

Essential revisions:– The role of odors in task performance. We would like to ask the authors to clarify their thinking and data with respect to the role of odors in the behavior. The manuscript as currently written gives the impression that odors play a negligible role, and that mouse behavior is comprised of a series of decisions internal to the brain (e.g., the use of the phrasing "sudden insight"; the abstract's statement that a mouse "executes correct 10-bit choices after only 10 reward experiences – a learning rate 1000-fold higher than in 2AFC experiments"; etc.) But it was not clear that odors play a negligible role.First, the mention of odors at the end of the discussion was a little confusing: although the text as written seems to suggest that the effect of odors was negligible, the data described seems to indicate the very opposite conclusion. The authors mention an experiment in which they rotate the maze, to ask whether animals are following a previously laid down odor trail, and they write that "Following that rotation the animal did in fact make a few visits to the rotated port location." Don't those first few visits to the rotated location suggest that indeed, the animal was following an odor trail? Why was behavior in those first few visits apparently discarded as inconsequential? It would be helpful if the authors clarified their thinking here. The authors proceeded to write that the mice "then quickly exploited the real water port again, without any apparent re-learning." But couldn't that same odor trail be used as a scaffold for rapid behavior change in runs subsequent to the first few ones?Second, the data for the odor experiments mentioned in those final discussion paragraphs should be shown and included as part of the manuscript (both the maze rotation data and the open water bowl data). Odors are a very significant potential confound to how these data are being interpreted. Allowing readers to see the data and evaluate it for themselves is important.

Regarding the water bowl experiments: These were done somewhat informally in an exploratory phase of the work on a different labyrinth. The revised text in Discussion omits their mention. There are stronger reasons to conclude that the water port does not attract the animals from a distance. For example the mice often turn away from the port even in close proximity, and discover it only after touching it. The reader can verify this directly in the raw videos.

Regarding the maze rotation experiments: We now have additional data and analysis from four animals and present these results in a new figure. The conclusion for all 4 animals: To navigate to the water port the animals do not depend on any cues that are attached to the maze. This includes any material they might have deposited, but also any construction details by which different locations in the maze might have been identified. Please see the new Figure 4 and accompanying text and methods.

Strictly speaking this conclusion applies only to the time point of the rotation, a few hours into the experiment, when the mice are experts at finding the water port. It is conceivable that the animal’s navigation policy changes in the course of learning, and the revised text includes this caveat. This and many other questions regarding the mechanisms of cognition will be taken up in separate experiments.

Third, we would like to invite the authors to consider and comment on the following and other odor-guided-behavior possibilities. This one seems, at first sight, to be consistent with much of their data. Consider a mouse that, as it runs it brushes against the walls, leaving behind a faint odor trail (it doesn't need to be urine or feces), and that tends to not reverse unless it reaches a dead end, but has a little bit of a tendency (doesn't need to be absolute) to avoid, based on odor, its own recent previous path. Wouldn't that mouse do a level 6 home run in its very first bout, with little overlap between entry and exit paths? Wouldn't it produce a trail very much like the trail shown in Figure 2A? Yet this mouse wouldn't have learnt anything internally, in its brain, about what constitutes a home path – the "learning" for this mouse would all have been in the physical odor trail it left behind.

Yes, that is an interesting proposal. A policy of "move to the location visited least recently and reverse at a cul-de-sac" would lead the mouse to perform a single complete scan of the labyrinth and then return from an end node. In practice this is not what happens during the animal’s first bout. We added a figure in the data repository with trajectories of the first bout for each of the 19 animals. In most cases the animal goes over some parts of the territory multiple times while leaving other regions untouched. On almost all the first home runs the mouse violates the rule to avoid its previous path, in that it enters a recently visited location rather than one that is still untouched. Of course, that doesn’t mean that odors play no role in the process.

That is only one possibility for how odor could affect behavior. There could be myriad others. After an hour, the maze is going to have feces and urine in multiple spots. As written, the claims about sudden insights, without considering whether such insight moments correlate with sudden odorant depositions (which would seem likely to act as highly prominent landmarks for the animal), could give readers the impression that the data demonstrate that it's all about internal mental processes. We are not yet convinced that odors can be so thoroughly ruled out.Fortunately, this can all be addressed in the writing. Whether or not mice use odors in the maze, the data is fascinating and the paper extremely interesting. We would invite the authors to consider addressing the issue from the beginning, in the introduction, so that readers can keep the issue in mind as they absorb the paper. (We phrase this as an invitation in the sense that this is a suggestion for the authors, not a mandatory requirement.) For example, a simple sentence in the introduction saying something like "preliminary control experiments suggested to us that odors did not play a major role in how mice learned and solved these navigation problems, but we did not fully control for odors and their potential role remains to be fully elucidated," would not detract from the work in any way, and would go a very long way towards making sure that, even as they are making their way through the paper, readers don't get the impression that the issue is settled and odors can be forgotten.

Yes, we now bring up the topic immediately after the section on rapid learning, which is when these questions will start forming in the reader’s mind. The maze rotation results are presented here. We also extended the topic in the discussion. The text acknowledges that the respective roles of olfaction and other senses remain to be worked out.

– We would ask the authors to consider the implications of having an imposed 90-second delay between rewards. The authors write that animals tend to explore the maze, as opposed to simply focusing on the water reward location (line 118; Figure 6 and accompanying main text). That the imposed 90-second delay could play a role in this is, surprisingly, not mentioned or considered.

Nothing in the experimental design forces the animal to leave the port during the timeout period; it could sit there and nap or groom and periodically test the port without penalty. We revised the text in Results and Discussion to state this explicitly. Fortunately the animals don’t adopt that lazy policy, otherwise it would be difficult to study the learning process. On the other hand, if we did not impose a timeout then the animal could drink to satiety on the first visit. Following that, it would likely explore the maze, just like all the animals that are fully sated to begin with. But we would again be deprived of the ability to watch the animal learn.

The 90-second delay also has an important implication for the "learning insight" analysis, as described in the next weakness below.– Another issue is in regard to a central claim in the manuscript, which is that animals have sudden insight moments in which they learn the path to the water reward. A plot based on data from one animal is presented in the main text (Figure 4B), appearing to the eye to support the claim (green and red lines have a kink in them). But for unexplained reasons data from other animals is relegated to the supplementary information (Figure 4 supplement 1), and many of these other animals do not, to the eye, support the claim (lines are smoothly curved, not kinked). The quantitative analysis that is used to adjudicate between sudden insights (a sudden step in the reward rate) versus gradual learning (slow continuous rise in reward rate) is flawed and, in our view, cannot be relied on: the approach compares a step model versus a smooth 2nd-order (quadratic) curve for how reward rate changes over time. However, because there is an imposed 90-second delay between successive rewards, the reward rate is capped at a maximum of 40 per hour, and cannot grow beyond that. A quadratic curve with a positive second-order term would rise indefinitely and will therefore be a poor fit to the data, not because the animals don't learn in a smooth way, but because of the 90-second delay, thus artifactually biasing the results of the analysis towards the step model.– We suggest that the authors show, in simulated data, how the 90 s delay would affect the quadratic ramp model to see how much of a straw man it is. While the authors are clear that they see discontinuities in only 5 out of the 10 water-deprived animals, perhaps it would be ideal for them to show a couple of the less-discrete learners in the main text for additional clarity.

Yes, the capped reward rate limits the range of reasonable parameters for the "ramp model". In response to this concern, we adopted a different analysis. We fit the rate of events with a sigmoid function having 4 parameters. It starts from a low rate and eventually saturates at some higher rate. In between it rises with an S-shape and the midpoint and width of the rise are again free parameters. The function fits all instances rather well, as shown in the revised supplementary figure.

The conclusions are almost identical to the previous approach: 6 of the 10 animals have a very short width parameter. We pursued these further to derive the uncertainty about the time of the step, as described in the revised methods. As recommended, we also moved one of the more gradual learners into the main text figure. Please see the revised Figure 5 with accompanying text and methods.

– "One shot learning." We don't think the authors have convincingly shown true one-shot homing behavior. It is an interesting and important claim in the manuscript ("one-shot learning of the home path" is the title of a section), but it is not yet completely clear that the data support that interpretation. The authors' point is that "home runs" start out longer than what random gradual exploration would predict, and that point is indeed supported by the data. But for the authors' claims about one-shot learning of home runs "from locations not previously encountered" to be supported, it is critical that the first and second bouts have little overlap between entry and exit paths. This should be shown in a main text figure, but it isn't, and moreover, Figure 5-supplement does not show initial individual runs, but rather averages over all bouts. Thus, the critical information about the first trajectories is nowhere in the current manuscript. We ask the authors to clearly show these initial-runs data for the individual mice. We do think Figure 2A suggests that their claim about path overlap in initial bouts may actually be true, but this should be demonstrated. Otherwise, we suggest the claim of one-shot learning be toned down.

We have added the requested analysis for each animal’s very first home run, and moved those figure panels to the main text as suggested. The conclusion remains the same. Please see the revised Figure 6 with accompanying text and methods.